# From local resynchronization to global pattern recovery in the zebrafish segmentation clock

**Koichiro Uriu[1†*], Bo-Kai Liao[2,3,4,5†], Andrew C Oates[3,4,5,6], Luis G Morelli[7,8,9]**

[1]Graduate School of Natural Science and Technology, Kanazawa University, Kakuma-machi, Kanazawa, Japan; [2]Department of Aquaculture, National Taiwan Ocean University, Keelung, Taiwan; [3]Department of Cell and Developmental Biology, University College London, Gower Street, London, United Kingdom; [4]The Francis Crick Institute, London, United Kingdom; [5]Max Planck Institute of Molecular Cell Biology and Genetics, Dresden, Germany; [6]Institute of Bioengineering, École polytechnique fédérale de Lausanne (EPFL), Lausanne, Switzerland; [7]Instituto de Investigación en Biomedicina de Buenos Aires (IBioBA) – CONICET – Partner Institute of the Max Planck Society, Polo Científico Tecnológico, Buenos Aires, Argentina; [8]Departamento de Física, FCEyN UBA, Ciudad Universitaria, Buenos Aires, Argentina; [9]Max Planck Institute for Molecular Physiology, Department of Systemic Cell Biology, Dortmund, Germany

*For correspondence:
uriu@staff.kanazawa-u.ac.jp

†These authors contributed equally to this work

Competing interests: The authors declare that no competing interests exist.

**Abstract** Integrity of rhythmic spatial gene expression patterns in the vertebrate segmentation clock requires local synchronization between neighboring cells by Delta-Notch signaling and its inhibition causes defective segment boundaries. Whether deformation of the oscillating tissue complements local synchronization during patterning and segment formation is not understood. We combine theory and experiment to investigate this question in the zebrafish segmentation clock. We remove a Notch inhibitor, allowing resynchronization, and analyze embryonic segment recovery. We observe unexpected intermingling of normal and defective segments, and capture this with a new model combining coupled oscillators and tissue mechanics. Intermingled segments are explained in the theory by advection of persistent phase vortices of oscillators. Experimentally observed changes in recovery patterns are predicted in the theory by temporal changes in tissue length and cell advection pattern. Thus, segmental pattern recovery occurs at two length and time scales: rapid local synchronization between neighboring cells, and the slower transport of the resulting patterns across the tissue through morphogenesis.

## Introduction

Synchronization of genetic oscillations in tissues generates robust biological clocks. To attain synchrony, cells interact with each other locally and adjust their phase of oscillations. How local interactions between oscillators lead to the emergence of collective rhythms has been studied in static tissues and in dynamic tissues with local cell rearrangements, but how collective rhythms are influenced by the more complex deformations of entire tissues typical in embryogenesis remains challenging and is less well understood. A system to explore this is the synchronization of genetic oscillations during the segmentation of the vertebrate embryo's body axis, a process termed somitogenesis. Cells in the unsegmented tissue, namely the presomitic mesoderm (PSM) and the tailbud, show collective rhythms of gene expression that set the timing of somite boundary formation and are referred to as the segmentation clock (*Oates et al., 2012*; *Pourquié, 2011*). In the tailbud,

spatially homogeneous oscillations can be observed. In the PSM, kinematic phase waves of gene expression move from posterior to anterior, indicating the presence of a spatial phase gradient along the axis (*Delaune et al., 2012*; *Shih et al., 2015*; *Soroldoni et al., 2014*). Importantly, this unsegmented oscillating tissue undergoes extensive deformations during the time of segment formation, with complex cellular rearrangements, flows and a changing global size and geometry (*Gomez et al., 2008*; *Jörg et al., 2015*; *Lawton et al., 2013*; *Mongera et al., 2018*; *Steventon et al., 2016*). However, our current understanding of synchronization in the segmentation clock follows largely from considering a non-deforming tissue with constant size.

In the zebrafish segmentation clock, Her1 and Her7 proteins repress their own transcription, forming negative feedback loops (*Hanisch et al., 2013*; *Schröter et al., 2012*; *Trofka et al., 2012*). These negative feedback loops are considered to generate cell-autonomous rhythms of gene expression (*Lewis, 2003*; *Monk, 2003*; *Webb et al., 2016*). Cells in the PSM interact with their neighbors via Delta-Notch signaling (*Horikawa et al., 2006*; *Jiang et al., 2000*; *Riedel-Kruse et al., 2007*; *Soza-Ried et al., 2014*). It is thought that Her proteins repress the transcription of *deltaC* mRNA, causing oscillatory expression of DeltaC protein on the cell surface (*Horikawa et al., 2006*; *Wright et al., 2011*). Binding of a Delta ligand to a Notch receptor expressed by neighboring cells leads to the cleavage and release of the Notch intracellular domain (NICD), which is translocated to the nucleus and modulates transcription of *her* mRNAs.

Several lines of evidence based on the desynchronization of the segmentation clock show that Delta-Notch signaling couples and thereby synchronizes neighboring genetic oscillators in the zebrafish PSM and tailbud. The first collective oscillation of the segmentation clock occurs immediately before the onset of gastrulation at 4.5 hr post fertilization (hpf), independently of Delta-Notch signaling (*Riedel-Kruse et al., 2007*; *Ishimatsu et al., 2010*). Thereafter, cells from embryos deficient in Delta-Notch signaling gradually become desynchronized due to the presence of various sources of noise (*Horikawa et al., 2006*; *Keskin et al., 2018*). Single-cell imaging of a live Her1 reporter in the Delta-Notch mutant embryos *deltaC/beamter*, *deltaD/after eight* and *notch1a/deadly seven* during posterior trunk formation (~15 hsspf) shows that Her1 protein oscillation is desynchronized across the PSM cells (*Delaune et al., 2012*). At the tissue level, Delta-Notch mutants form the anterior 4 ~ 6 segments normally, followed by consecutive defective segments (*van Eeden et al., 1996*). These phenotypes are not caused by a direct failure of segment boundary formation (*Ozbudak and Lewis, 2008*), but have been explained in terms of the underlying desynchronization of the segmentation clock (*Jiang et al., 2000*; *Riedel-Kruse et al., 2007*).

This desynchronization hypothesis has been formalized as a theory based on coupled oscillators (*Riedel-Kruse et al., 2007*; *Liao et al., 2016*). The theory postulates a critical value $Z_c$ such that if the level of synchrony becomes lower than this critical value, a defective segment boundary is formed, *Figure 1A*. In the absence of Delta-Notch signaling, the level of synchrony decays over time and eventually becomes lower than $Z_c$. The time point at which the level of synchrony crosses $Z_c$ for the first time is considered to set the anterior limit of defects (ALD), that is the anterior-most defective segment along the body axis. Indeed, theory based on this desynchronization hypothesis can quantitatively explain the ALD in Delta-Notch mutants (*Riedel-Kruse et al., 2007*).

The desynchronization hypothesis implies that the oscillators could be resynchronized by restoring Delta-Notch signaling, with the expectation that resynchronization of the segmentation clock requires several oscillation cycles for the level of synchrony to smoothly increase and surpass the threshold $Z_c$, giving rise to a transition from defective to recovered normal segments. Due to the constitutive lack of coupling, Delta-Notch mutants cannot be used to analyze resynchronization dynamics. A powerful tool for this purpose is the Notch signal inhibitor DAPT, which inhibits the cleavage and release of NICD, blocking cell coupling. Importantly, DAPT can be washed out to recover cell coupling and this allows cells to resynchronize their genetic oscillations, (*Figure 1A*; *Riedel-Kruse et al., 2007*; *Ozbudak and Lewis, 2008*; *Liao et al., 2016*; *Mara et al., 2007*). Previous experimental studies showed that after late washout of DAPT at the nine somite stage, embryos start making normal segments again after several oscillation cycles (*Riedel-Kruse et al., 2007*; *Liao et al., 2016*; *Mara et al., 2007*). The position of the first recovered normal segment after DAPT washout represents the time at which the level of synchrony surpasses $Z_c$ for the first time (*Liao et al., 2016*). In previous studies, almost all segments formed normally after the first fully recovered segment (*Riedel-Kruse et al., 2007*; *Liao et al., 2016*), consistent with a monotonic increase of the level of synchrony in the vicinity of $Z_c$, *Figure 1A*, as expected from the

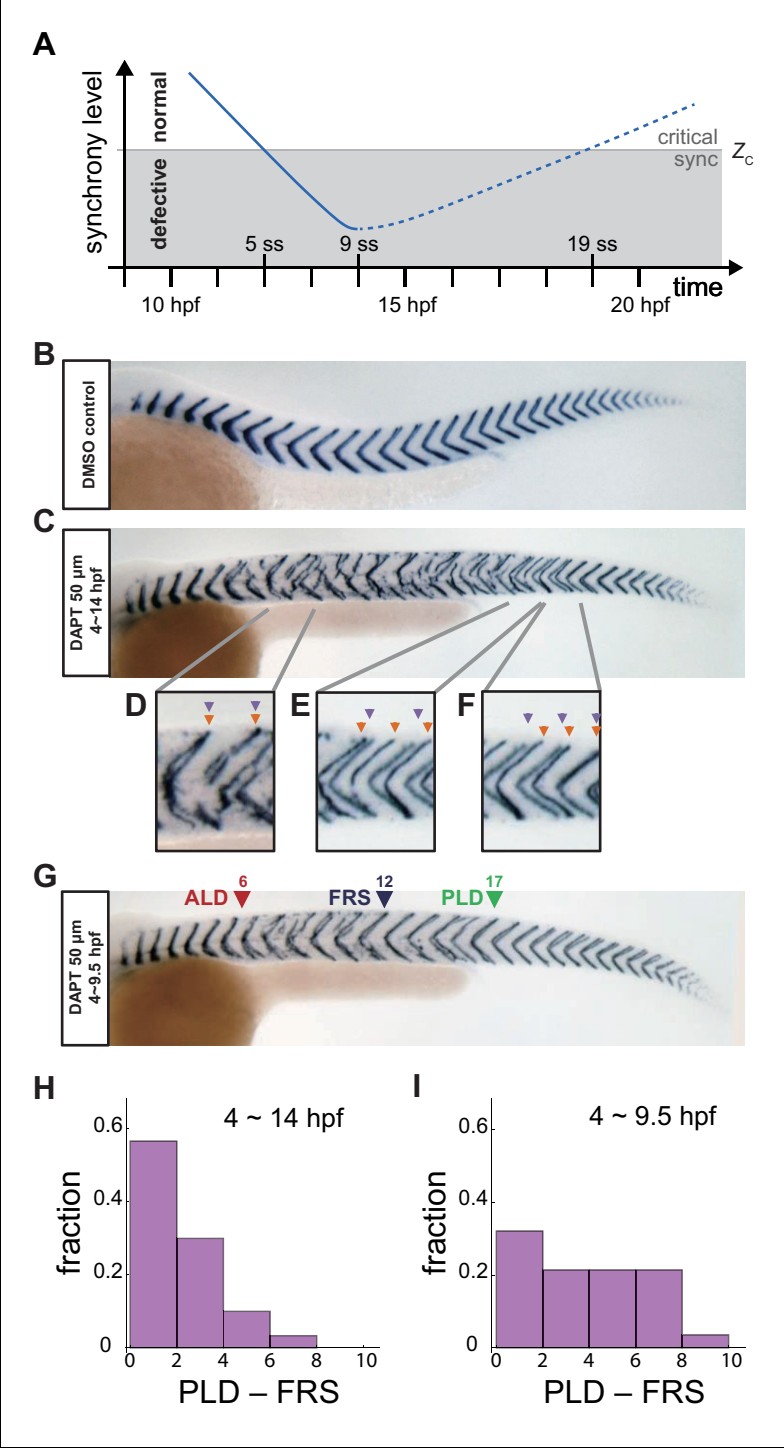

**Figure 1.** Segment boundary defects observed in late and early DAPT washout embryos. (**A**) Schematic time series of synchrony level during desynchronization and resynchronization. In the presence of DAPT, the synchrony level decreases due to the loss of Delta-Notch signaling (solid line). DAPT is washed out at 14 hr post-fertilization (hpf; ~9 somite stage; ss) in this panel and resynchronization starts from that time point (dotted line). If the synchrony level is higher (lower) than a critical value $Z_c$, normal (defective) segments are formed. (**B**) Wild-type control embryo treated with DMSO. (**C**) Embryo with late DAPT washout at 14 hpf (9 ss). Enlargements of (**D**) broken or fragmented boundaries, (**E**) incorrect number of boundaries and (**F**) left-right misaligned boundaries are shown below. (**G**) Embryo with early DAPT washout at 9.5 hpf (0 ss). Red, blue and green triangles indicate the anterior limit of defect (ALD), first recovered segment (FRS) and posterior limit of defect (PLD), respectively. (**H**), (**I**)

*Figure 1 continued on next page*

*Figure 1 continued*

Histograms of the difference between PLD and FRS (PLD – FRS) for embryos with DAPT washout at (**H**) late (14 hpf; *n* = 30) and (**I**) early (9.5 hpf; *n* = 28) stages. Numbers of embryos examined in (**H**) and (**I**) were 15 and 14, respectively. FRS and PLD were measured separately between left and right sides of embryos. p<0.05 in Kolmogorov-Smirnov test.

The online version of this article includes the following source data and figure supplement(s) for figure 1:

**Source data 1.** Segment boundary defects in embryos with different DAPT washout timing.

**Figure supplement 1.** Difference between FRS and PLD in experimental data.

desynchronization hypothesis. Despite this success, it remains fundamentally unclear how tissue-scale gene expression patterns underlying segment recovery reorganize from local intercellular interactions and whether they are affected by tissue size and shape changes that occur during development.

Here, we analyze resynchronization dynamics of the zebrafish segmentation clock at different developmental times using both experimental and theoretical techniques. In contrast to late washout mentioned above, we find that washing out DAPT at earlier developmental stages causes a region of scattered segment defects, where normal and defective segments are intermingled. This striking phenotype was not anticipated by previous models (*Riedel-Kruse et al., 2007*; *Uriu et al., 2017*). To investigate the processes in the segmentation clock that yield this distinctive recovery behavior, here we develop a new model of the segmentation clock that encompasses two scales, describing resynchronization and pattern recovery in terms of local interactions between cells in the tissue, as well as global properties of the tissue. In concert, we develop observables that allow pattern dynamics to be quantified and compared between simulation and experiment, the vorticity index and local order parameter. Despite its simplicity, this model can describe the intermingling of normal and defective segments. Numerical simulations indicate that persistent phase vortices appear during resynchronization, resulting in scattered and intermingled segment defects along the axis. The length of the PSM and tailbud and advection pattern influence the recovery process via the transport of phase vortices from the posterior to the anterior of the PSM. Moreover, by including temporal changes to tissue length, advection pattern and coupling strength, the model makes predictions about the pattern of resynchronization at both early and late stages that we confirm experimentally in the embryo.

## Results

### Scattered embryonic defective segments in zebrafish resynchronization assay

To investigate the processes involved in resynchronization of the segmentation clock, we used a resynchronization assay based on washing out the Notch signaling inhibitor DAPT at different developmental stages. In this assay, zebrafish embryos were placed in DAPT for a defined duration, during which time the segmentation clock desynchronized and defective segments were formed, then washed extensively to allow Delta-Notch signaling activity to resume. Subsequently, the segmentation clock gradually resynchronized and normal segments were made.

Throughout this study, we administered DAPT at 4 hpf, a developmental stage before the oscillating genes of the segmentation clock were expressed. This was a treatment duration of at least 5 hr, sufficient to obtain defects on both left and right sides of the embryo for subsequent resynchronization analysis (*Ozbudak and Lewis, 2008*). A record of the resulting spatiotemporal pattern of somitogenesis was visualized after its completion by whole-mount in situ hybridization for the myotome segment boundary marker gene *xirp2a* (*xin actin-binding repeat containing 2a*) in ~36 hpf embryos, *Figure 1B,C*. This assay was chosen for its high sensitivity for boundary defect detection (*Riedel-Kruse et al., 2007*), and we analyzed these staining patterns by scoring boundaries as either normal or defective using established criteria (*Riedel-Kruse et al., 2007*; *Liao et al., 2016*), with the exception that we scored the left and right embryonic sides separately. Examples of defects observed with DAPT treatment included fragmented, mis-spaced, or mis-aligned boundaries, *Figure 1D–F*. To prevent incorrect identification of misaligned boundaries in embryos with bent

axes, or in tilted samples, we confirmed that the boundaries outside of the defective region were well aligned between left and right sides. To assign the ordinal segment number to defective boundaries when boundaries were severely fragmented, we used the contralateral side or counted either dorsal or ventral boundary ends, which were often clearer, to estimate their axial position.

As described in the introduction, the location of the transition from normal to defective segments resulting from desynchronization is termed the anterior limit of defects (ALD), given by the first segment along the embryo's axis that shows a defective boundary, *Figure 1—figure supplement 1A*. After removing DAPT, resynchronization begins and normal segments form eventually. This transition has been recorded by the first recovered segment along the axis (FRS) (*Liao et al., 2016*) and then the posterior limit of defects (PLD), the most posterior segment along the axis with a boundary defect, *Figure 1—figure supplement 1A* (*Riedel-Kruse et al., 2007*). Note that because segments form rhythmically and sequentially along the body axis, FRS and PLD label both an axial position and the developmental time of segment formation.

In late washout experiments, we observed that a normal segment boundary sometimes formed shortly after ALD even when DAPT was still present, possibly due to desynchronization fluctuations. In previous reports, the definition of the FRS avoided counting these early defects because washout was done late, after full desynchronization. However, when DAPT was washed out early, before ALD occurred, we could not discriminate whether a normal segment formed due to desynchronization fluctuations or as a consequence of resynchronization. The frequency of defects kept growing during an early phase in all conditions until reaching a plateau around segment 9, *Figure 1—figure supplement 1B*, suggesting that the desynchronization phase lasted until segment 9, at least. Hence, in this study we defined FRS as the first recovered segment after segment 9, when the desynchronization phase was over.

We first analyzed the recovery of normal segments when DAPT was washed out at 14 hpf ($t_{wash-out}$ = ~9 somite-stage (ss)), as in previous studies, *Figure 1C*. Several defective segment boundaries were formed after washout, suggesting that the level of synchrony was still lower than the critical value for normal segment formation during that time interval. However, embryos recovered a normal segment boundary after some time, indicating that the level of synchrony surpassed the threshold, *Figure 1C*. With this late washout time, we often observed contiguous defective segments before FRS, suggesting that cells in the PSM were completely desynchronized by a DAPT pulse of this length. In addition, PLD coincided closely with FRS, with the distribution of PLD – FRS peaking at lowest values, *Figure 1H*, *Figure 1—figure supplement 1A,F*, suggesting that once the level of synchrony surpassed the critical value $Z_c$, it remained larger than $Z_c$, as expected. This observation can be interpreted using the desynchronization hypothesis to indicate that the level of synchrony increases monotonically over time, resulting in the formation of consecutive normal segments after the FRS, *Figure 1A*.

Importantly, however, when we washed DAPT out at an earlier time $t_{wash-out}$ = 9.5 hpf (~0 ss), the majority of embryos had an interval along the axis where normal and defective segments were intermingled between FRS and PLD, with the difference between PLD and FRS distributed more uniformly, *Figure 1G,I*, and *Figure 1—figure supplement 1A,C*. This result suggests that the segmentation clock has a level of synchrony close to $Z_c$ in this intermingled region, and persistent fluctuations in synchrony level lead to intermittent defective boundary formation.

## Physical model of the PSM

According to the desynchronization hypothesis, intermingling of normal and defective segment boundaries suggests a fluctuation of the phase order parameter around its critical value for proper segment formation. How could such large and potentially long-lasting fluctuations of the phase order occur? The desynchronization hypothesis (*Jiang et al., 2000*) was first formalized in a mean-field theory describing synchronization dynamics from global interactions (*Riedel-Kruse et al., 2007*). Later, synchronization in the tailbud was analyzed with a theory with local interactions and neighbor exchange by cell mobility (*Uriu et al., 2017*; *Uriu et al., 2010*). However, a critical prediction of these theories is that once a population of oscillators is synchronized, a large fluctuation of synchrony level is not expected (*Hildebrand et al., 2007*; *Kuramoto and Nishikawa, 1987*; *Daido, 1987*). Instead of such global phase order fluctuations, other hypotheses for the intermingled defects are the emergence of localized disorder, or the existence of local phase order with a mis-orientation to the global pattern. To explore these potential behaviors, following the general lineage

of the clock and wavefront (*Cooke and Zeeman, 1976*), we develop a physical model of the PSM and tailbud that brings together in a novel framework previous descriptions of (i) the local processes of phase coupling (*Morelli et al., 2009*) and physical forces (*Uriu et al., 2017*; *Uriu and Morelli, 2014*) between neighboring oscillators, and (ii) the tissue-level properties of a frequency profile and oscillator arrest front (*Jörg et al., 2015*; *Morelli et al., 2009*), changing tissue length (*Jörg et al., 2015*), and a gradient of cell mixing (*Uriu et al., 2017*); furthermore, we introduce an advection pattern of the PSM (*Jörg et al., 2015*) that changes in time, *Figure 2*, *Figure 2—figure supplement 1* and *Supplementary file 1*.

The model describes the PSM and tailbud as a U-shaped domain in a 3D space, *Figure 2A*, *Figure 2—figure supplement 1*, see Materials and methods for details. We set the posterior tip of the

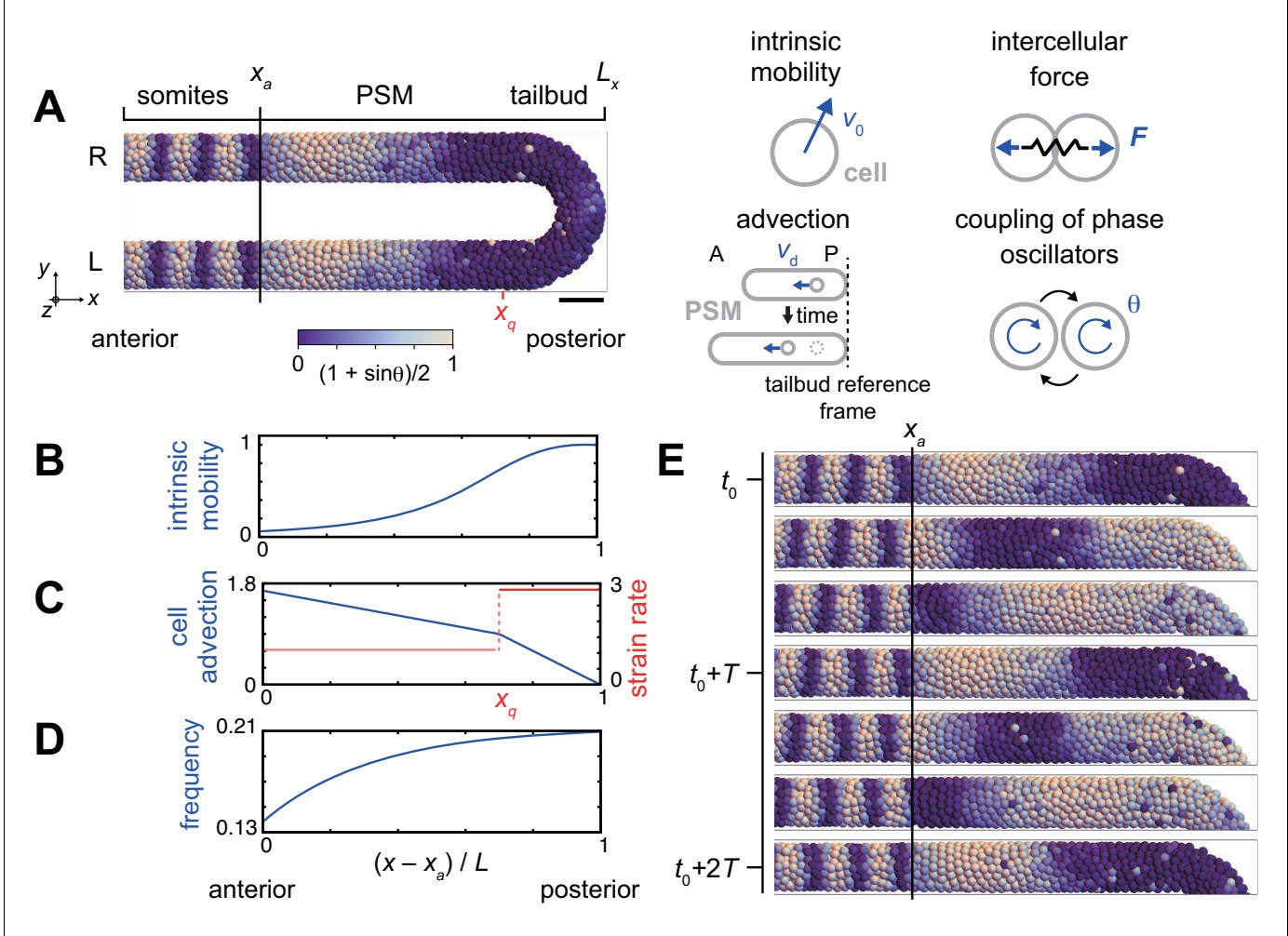

**Figure 2.** Physical model of the PSM and tailbud. (A) U-shape geometry of the PSM and tailbud (left), and schematics of key ingredients in the model (right). Each sphere represents a PSM cell. The scale bar indicates the mapping of phase $\theta_i$ to color: white is $\pi/2$ and blue is $3\pi/2$. R: right. L: left. Scale bar: 50 μm. (B) Intrinsic cell mobility gradient, (C) cell advection speed, and (D) autonomous frequency gradient along the anterior-posterior axis of the PSM and tailbud. In (C), the absolute value of the spatial derivative of advection speed, referred to as strain rate, is indicated by the red line. $L$ is the length of the PSM $L = L_x - x_a$. (E) Kinematic phase waves moving from the posterior to anterior PSM in a simulation. Snapshots of the right PSM are shown. See also *Figure 2—video 1*. $t_0 = 302$ min is a reference time point. $T = 30$ min is the period of oscillation at the posterior tip of the tailbud. Parameter values for simulations are listed in *Supplementary file 1*.

The online version of this article includes the following video and figure supplement(s) for figure 2:

**Figure supplement 1.** 3D geometry of the PSM and tailbud in the physical model (A) Two tubes and a half torus represent the PSM and tailbud, respectively with anterior to the left and posterior to the right.

**Figure 2—video 1.** Traveling phase waves in a constant tissue.
https://elifesciences.org/articles/61358#fig2video1

tailbud as a reference point. Cells are represented as particles in the 3D space, subject to physical forces from other particles when they are closer than a typical length scale that we term cell diameter. In addition, tissue boundaries exert confinement forces on cells (*Uriu et al., 2017*). Although cells are rendered as spheres in simulations, their effective shapes are in fact dynamic and depend on their local physical interactions. In accordance with previous experimental studies, we consider the spatial gradient of intrinsic cell mobility across the PSM and tailbud, with highest mobility in the posterior, *Figures 2B* (*Lawton et al., 2013*; *Mongera et al., 2018*; *Uriu et al., 2017*). Axis extension as observed in the lab is described here, from the reference point of the posterior tip of the tailbud, as cell advection from posterior to anterior, *Figures 2A,C* (*Jörg et al., 2015*; *Morelli et al., 2009*; *Ares et al., 2012*). The value of the spatial derivative of the advection velocity at each position effectively represents the local strain rate, *Figure 2C*. Cellular motion is described by the overdamped equation with four terms that represent the four physical influences listed above:

$$\frac{d\mathbf{x}_i}{dt} = \mathbf{v}_d(\mathbf{x}_i) + v_0(\mathbf{x}_i)\mathbf{n}_i(t) + \sum_{j=1,j\neq i}^{N} \mathbf{F}(\mathbf{x}_i, \mathbf{x}_j) + \mathbf{F}_b(\mathbf{x}_i),$$

where $\mathbf{x}_i$ is the position of cell $i$, $\mathbf{v}_d(\mathbf{x}_i)$ is cell advection velocity, $v_0(\mathbf{x}_i)$ is the speed of intrinsic cell movement with direction $\mathbf{n}_i(t)$, $\mathbf{F}(\mathbf{x}_i,\mathbf{x}_j)$ is the physical contact force between cells $i$ and $j$, $N$ is the total cell number and $\mathbf{F}_b(\mathbf{x}_i)$ is the boundary force that confines cells within the U-shaped domain.

Genetic oscillation in each PSM cell is described as a phase oscillator with noise terms. The phase oscillators are coupled with their neighbors, representing intercellular interaction with Delta-Notch signaling. We define cells to be neighbors when their distance is shorter than the cell diameter. We also consider a left-right symmetric frequency profile along the anterior-posterior axis of the PSM, as observed in zebrafish embryos, to create kinematic phase waves (*Figure 2D*; *Shih et al., 2015*; *Soroldoni et al., 2014*; *Jörg et al., 2015*). The mobility of cells in the tailbud increases the communication of the phase between left and right halves. The frequency of oscillation is highest at the tip of the tailbud and gradually decreases toward the anterior PSM. The phase equation describing the oscillators has three terms, representing the frequency profile of the cell-autonomous oscillators, the coupling between neighbors, and noise:

$$\frac{d\theta_i(t)}{dt} = \omega(\mathbf{x}_i) + \frac{\kappa(t)}{n_{\theta i}(t)}\sum_{|\mathbf{x}_j-\mathbf{x}_i|\leq d_c}\sin\left[\theta_j(t)-\theta_i(t)\right] + \sqrt{2D_\theta}\xi_{\theta i}(t),$$

where $\theta_i$ is the phase of cell $i$, $\omega(\mathbf{x}_i)$ is the autonomous frequency at position $\mathbf{x}_i$, $\kappa$ is the coupling strength, $n_{\theta i}$ is the number of neighboring cells interacting with cell $i$, $d_c$ is the cell diameter, $D_\theta$ is the phase noise intensity and $\xi_{\theta i}(t)$ is a white Gaussian noise. When a simulation was started from a uniformly synchronized initial condition, the frequency profile generated left-right symmetrical kinematic phase waves due to growing phase differences between the anterior and posterior PSM, *Figure 2E* and *Figure 2—video 1*.

To model the formed segments, we arrest the oscillation when cells leave the PSM from its anterior end $x_a$, *Figure 2A,E*. The arrested phase stripes in the region $x < x_a$ are representative of segment boundaries, and the segment length is the wavelength of this arrested phase pattern. Although the determination of the segment boundary in vivo is a complex process (*Dahmann et al., 2011*; *Naganathan and Oates, 2020*), for simplicity we consider that these phase stripes correspond to the boundaries of the resulting morphological segments and the chevron-shaped myotomes that are detected in our experiments. Cells in the formed segment region $x < x_a$ continue to be advected anteriorly at the same speed as the anterior end of the PSM, due to axial elongation, and their relative positions are fixed. While small heterogeneities in cell density and cell division may exist in the tissue (*Mongera et al., 2018*; *Steventon et al., 2016*; *Uriu et al., 2017*; *Zhang et al., 2008*), as a simplifying assumption we keep global cell density constant over time. Consequently, we add new cells with random phase values (*Horikawa et al., 2006*) at random positions in the PSM and tailbud to balance the cells exiting through the anterior end of the PSM.

## Simulation results 1. Intermingled defective segments result from spatially heterogeneous resynchronization in the PSM

Using this physical model, we analyzed resynchronization dynamics in simulations. As an initial condition, we described the state of the PSM and tailbud immediately after DAPT washout by assigning random initial phases to cells, top panel in *Figure 3A*, as an extended treatment of saturating dose of DAPT was expected to cause such complete randomization of oscillator phases (*Delaune et al., 2012*). In this desynchronized state, normal segment boundaries – as defined by ordered phase stripes – did not form, matching the experimental appearance of embryos with persistent loss of Delta-Notch signaling. For simplicity, we start our analysis with constant tissue parameters. Below, we will introduce temporal changes to the parameters.

*Figure 3A* shows snapshots of a synchrony recovery simulation, see also *Figure 3—video 1*. To characterize resynchronization dynamics, we computed local phase order at each position along the anterior-posterior axis, *Figure 3B* and *Figure 3—figure supplement 1*. Due to local coupling, cells first formed local phase synchronization, *Figure 3A,B* and *Figure 3—video 1*. During the first stage of this synchrony recovery, the kymograph of local phase order shows three locally synchronized domains that extended their size due to an increase in number of synchronized cells at the same time as they were advected in an anterior direction, *Figure 3B,E*. When the size of the most anterior domain with local phase order above a threshold of 0.85 ($Z_c$) exceeded one segment length at its arrival in the anterior end of the PSM, a first recovered segment (FRS) was formed in a simulation, blue triangles in *Figure 3A,B,F*. However, because local interactions drove resynchronization in a spatially heterogeneous manner, domains where phase order was lower than $Z_c$ could exist more posterior to such a well-synchronized domain, *Figure 3B,E*. Subsequent arrival of the less synchronized domains caused fluctuation of the local order parameter in the anterior end of the PSM, which resulted in defective segment formation after FRS, *Figure 3A,F*. Note that patterns of synchronized domains in the left and right sides of the PSM were not well correlated during this time interval (*Figure 3—video 1*) despite a left-right symmetrical frequency profile. This is in contrast to the fully recovered state, where in the presence of well-synchronized oscillators the symmetrical frequency profile and the communication of phase through the tailbud ensures the left-right symmetry of the wave pattern. Thus, the numerical simulation suggests that a sequence of synchronized and less synchronized domains moving along the PSM results in an intermingling of normal and defective segments along the axis. This intermingling matches the experimental observations.

These less-synchronized domains typically formed persistent phase patterns that rotated along an axis as a vortex, as illustrated in the simulation, *Figure 3D* and *Figure 3—video 1*. To detect these patterns, we introduced an index referred to as vorticity, *Figure 3—figure supplement 2* and Materials and methods. In brief, the vorticity index detects the core of a vortex, where the phase values circulate from 0 to $2\pi$ around a point, but does not measure the spatial extent of the vortex. The kymograph of the vorticity indicates that the less synchronized domains in the kymograph of phase order were caused by the phase vortices, *Figure 3A–C*. When a vortex was brought to the anterior end of the PSM through cell advection, it was converted into a defective segment boundary and delayed the PLD, *Figure 3B,C*. Although this process is best visualized dynamically in simulations, one example is illustrated for the vortex in *Figure 3D* and the resulting local phase order and a segment defect in *Figure 3F* (bracket). The formation of this particular segmental defect corresponds to time points 350-380 min on the right hand side in *Figure 3—video 1*. Thus, the formation of persistent local phase patterns with a mis-orientation to the global pattern in the posterior PSM caused defective segment boundaries after the FRS, providing an explanation for the early washout experiments.

The passage of each vortex into the anterior PSM in the simulations generated a segment defect with a characteristic length of one or two segments, *Figure 3A–D,F*. We compared these defect lengths between simulation and embryonic data and found that across all embryonic stages (*Figure 1—figure supplement 1*) and all simulations (see Figure 4, below), the size distributions were in quantitative agreement, *Figure 3—figure supplement 3*. The resulting invariant length scale of the defect size is shown for simulation and embryonic data in *Figure 3G,H*. This finding is consistent with phase vortices as the origins of intermingled segment defects in the embryo.

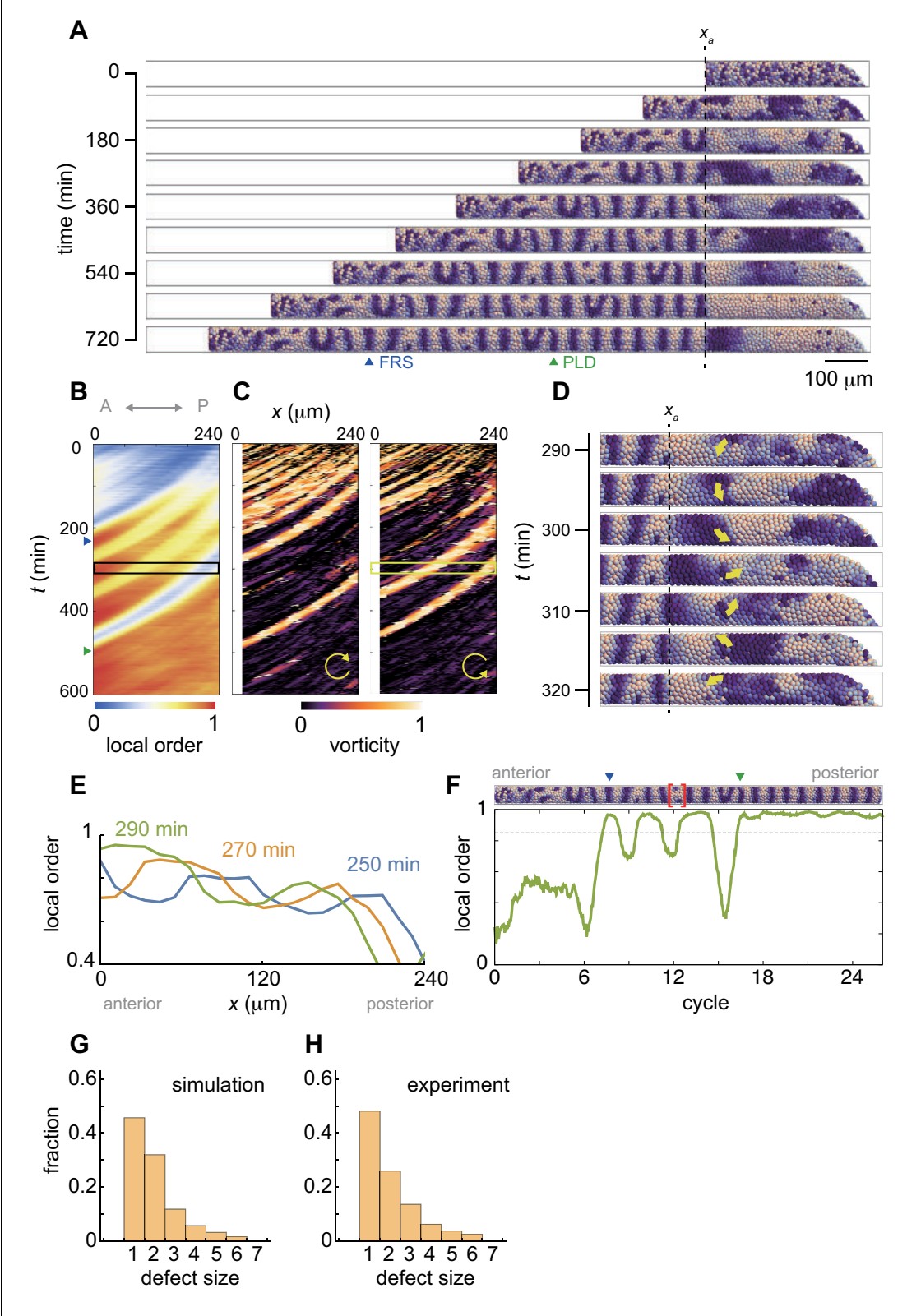

**Figure 3.** Resynchronization simulations with constant tissue parameters. (A) Snapshots of a resynchronization simulation. Color scale as in *Figure 2A*, also in (D) and (F). The black dotted vertical line indicates the position of the anterior end of the PSM $x_a = 0$. Tissue parameters are constant over time. See also *Figure 3—video 1*. (B-F) Analysis of local phase order and vortex transport in the simulation shown in (A). (B) Kymograph of local phase order parameter of the right PSM shown in (A). (C) Kymographs of phase vorticity for (left) clockwise and (right) counter clockwise rotations. The phase

*Figure 3 continued on next page*

*Figure 3 continued*

patterns within the black and yellow boxed space-time domains in (B) and (C) are shown in (D). (D) Snapshots of a phase vortex. The yellow arrows indicate the direction of rotation. (E) Local phase order parameters along the anterior-posterior axis of the PSM at different time points. (F) Time series of local phase order parameter at the anterior end of the right PSM $x_a$. The horizontal broken line indicates the threshold $Z_c = 0.85$ for determining normal and defective segments in simulations. The resultant stripe pattern is on top. In (A), (B) and (F), the blue and green triangles mark FRS and PLD, respectively. Red bracket in (F) highlights a segmental defect resulting from vortex in (D). Parameter values for simulations are listed in *Supplementary file 1*. (G), (H) Defect size distributions for (G) simulation ($n = 800$) and (H) embryonic experimental data ($n = 134$). Defect size indicates how many consecutive segment boundaries are defective in between FRS and PLD. In (G) and (H), the data for different washout timing shown in *Figure 4* and *Figure 1—figure supplement 1* were pooled to make the histograms. The defect size distribution for each washout timing is shown in *Figure 3—figure supplement 3*.

The online version of this article includes the following video and figure supplement(s) for figure 3:

**Figure supplement 1.** Definitions of local phase order and a normal segment boundary in simulations.

**Figure supplement 2.** Calculation of vorticity.

**Figure supplement 3.** Typical defect size caused by phase vortices in simulation is consistent with the one observed in embryonic experimental data.

**Figure supplement 4.** Faster cell mixing reduces PLD whereas it does not influence FRS.

**Figure supplement 5.** Dependence of FRS and PLD on the PSM length in simulations.

**Figure supplement 6.** Dependence of FRS and PLD on PSM advection pattern in simulations.

**Figure supplement 7.** PSM radius $r$ influences both FRS and PLD.

**Figure supplement 8.** Coupling strength $\kappa_0$ influences both FRS and PLD.

**Figure supplement 9.** Dependence of desynchronization and resynchronization on phase noise intensity $D_\theta$.

**Figure supplement 10.** Weak dependence of FRS and PLD on the shape parameter $k$ of the frequency profile.

**Figure supplement 11.** Weak dependence of FRS and PLD on the torus radius for the tailbud $R$.

**Figure 3—video 1.** Resynchronization simulation in a constant PSM tissue.

https://elifesciences.org/articles/61358#fig3video1

## Simulation results 2. Dependence of FRS and PLD on each tissue parameter

These results show that the model captures the intermingling of normal and defective boundaries frequently observed in the early washout experiments, but can the model also capture the axial distribution of FRS and PLD observed in the late washout experiments, thereby joining these observations into a coherent picture of resynchronization across developmental stages?

The finding that transport of local phase patterns across the tissue can influence segment recovery suggests that in addition to local intercellular interactions, tissue level parameters may also be important. Previous experimental studies showed that the PSM length becomes shorter as segments are added (*Soroldoni et al., 2014*; *Gomez et al., 2008*). Convergent extension by cells in the anterior part of the tissue contributes to advection pattern in the PSM at early developmental stages (*Yin et al., 2008*). At later developmental stages, cellular flows from the tissues dorsal to the tailbud change the advection pattern (*Lawton et al., 2013*; *Mongera et al., 2018*; *Steventon et al., 2016*). These complex rearrangements are represented in our model in a simplified manner by the advection profile. Thus, several lines of experimental data support changes in the PSM length and its advection pattern, and other properties may also vary during development. We therefore studied how the FRS and PLD depend on each of the tissue parameters in the physical model. We begin by shifting a given single parameter to a new constant value, while leaving the others unchanged for the simulation, *Table 1* and *Figure 3—figure supplements 4–11*, and return to the time-dependent cases in the next section. Simulations were started from complete random initial phases as in *Figure 3*. We computed FRS and PLD over 100 different realizations of simulations and the results are summarized in *Table 1*. See Materials and methods for the quantification of FRS and PLD in simulations.

We found that the speed of cell mixing at the tailbud, PSM length $L$ and the cell advection pattern in the PSM did not affect FRS, whereas they strongly affected PLD, *Figure 3—figure supplements 4–6*. Faster cell mixing in the tailbud reduced PLD, *Figure 3—figure supplement 4*, because it prevented the formation of persistent phase vortices at the tailbud. The PSM length and cell advection pattern determined the time taken for a phase vortex to reach the anterior end of the PSM. Longer PSM increased PLD because a phase vortex needed more time to arrive at the anterior end by the cell advection, *Figure 3—figure supplement 5*. Cell advection underlies the transport of

**Table 1.** Dependence of FRS and PLD on each tissue parameter.
These results were obtained with simulations where all the tissue parameters were constant over time.

| Influence | Tissue parameters |
| --- | --- |
| Change in only PLD | PSM length, cell mixing, advection pattern |
| Change in only FRS | None |
| Change in both FRS and PLD | Coupling strength, PSM radius (tube radius) |
| No or weak effect | Frequency profile, tailbud size (torus radius), phase noise intensity |

phase vortices, *Figure 2C* and *Figure 3—figure supplement 6*. If cell advection was slower at the posterior part of the PSM, PLD became larger because vortices stayed longer at the posterior part, *Figure 3—figure supplement 6B*. In contrast, if cell advection was faster at the posterior part, PLD became smaller due to faster transport of phase vortices to the anterior part, *Figure 3—figure supplement 6B*. Thus, these parameters are important for understanding the new phenotype, because each can increase the difference between FRS and PLD, producing the intermingled defects observed experimentally.

In contrast, the PSM radius $r$ and the coupling strength between neighboring oscillators $\kappa_0$ influenced both FRS and PLD, *Figure 3—figure supplements 7* and *8*. Smaller PSM radius $r$ decreased FRS and PLD, *Figure 3—figure supplement 7*. Smaller cell number at each position in the PSM with a smaller radius $r$ allowed local coupling to more rapidly generate a synchronized domain as large as the tissue diameter, leading to a normal segment. Larger values of $\kappa_0$ reduced FRS and PLD, *Figure 3—figure supplement 8*. A larger coupling strength reduced the time for local order to form, including vortex patterns. As a result, the last-formed vortex departed the posterior PSM earlier, decreasing PLD.

Coupling keeps phase differences between neighboring oscillators in check. There are two sources of local phase fluctuations in the model: (i) the noise term in individual phase dynamics and (ii) the addition of new cells with random phase values. Desynchronization simulations, where the coupling between cells was absent, demonstrated that the addition of new cells alone was enough to disrupt the wave pattern and compromise the integrity of segment boundaries, *Figure 3—figure supplement 9*. Additionally, a larger phase noise intensity further contributed to a faster decay of the pattern. However, in resynchronization simulations with coupling, both FRS and PLD only weakly depended on the noise intensity for biologically realistic values, *Figure 3—figure supplement 9*.

Finally, we found that the shape of frequency profile and the torus size for the tailbud $R$ did not influence either FRS or PLD, *Figure 3—figure supplements 10* and *11*. Note that there was no parameter that influenced only FRS, *Table 1*. In summary, FRS was determined by parameters that influence local synchronization of oscillators. PLD, on the other hand, was influenced by parameters that control local synchronization, formation of phase vortices, and their arrival at the anterior end of the PSM.

## Simulation results 3. Prediction of PLD from DAPT washout timing, PSM shortening and changing advection pattern

In the previous section, we considered constant values of parameters defining coupling and tissue properties. However, as noted above, some features like PSM length vary during development on timescales that may be relevant for resynchronization (*Soroldoni et al., 2014*; *Gomez et al., 2008*; *Jörg et al., 2015*). To further investigate whether the early and late segmentation phenotypes shown in *Figure 1* could result from a common underlying set of processes, we introduced the washout process into the model, and examined the effect of different washout times in simulations in which tissue properties changed over the course of the simulation.

To model differences in timing of DAPT washout, we started with coupling strength $\kappa(t) = 0$ for $t < t_{wash-out}$ and then switched on coupling at $t = t_{wash-out}$. We assigned random phases to the oscillators in the model as an initial condition, assuming that all DAPT treatments completely desynchronized oscillators, as above. Hence, the phase disordered state lasted until $t = t_{wash-out}$ and resynchronization begun at that time. We performed 100 realizations of simulations for each washout

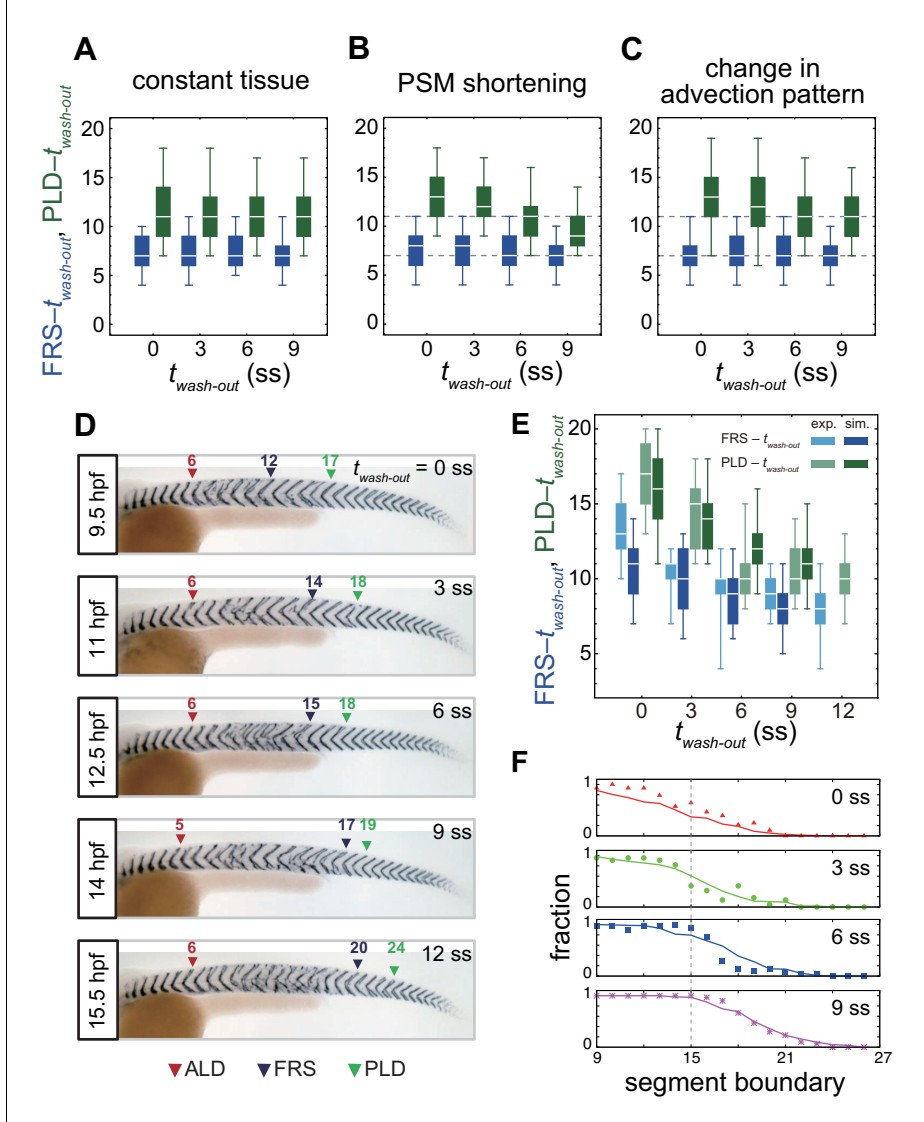

**Figure 4.** Gradual transition from early to late washout boundary phenotypes captured by the physical model. (A-C) Dependence of times to FRS and PLD on DAPT washout time for different conditions in simulations. (A) Constant tissue where all the tissue parameters remain unchanged during a simulation. (B) PSM length becomes shorter with time. All the other parameters are constant. See also *Figure 4—video 1*. (C) Cell advection pattern changes at 9 somite stage (ss). Before 9 ss, the strain rate is larger in the anterior than posterior PSM. After 9 ss, the strain rate becomes larger in the posterior PSM. See also *Figure 4—video 2*. All the other parameters are constant. The box-whisker plots indicate 5, 25, 75, and 95 percentiles. The white bars mark the median. In (B) and (C), the gray dotted lines mark the medians of FRS and PLD in the constant tissue shown in (A). (D) Whole-mount in situ hybridization for the myotome segment boundary marker gene *xirp2a* in ~36 hr post-fertilization (hpf) embryos. DAPT washout time is 9.5 hpf (0 ss; *n* = 28), 11 hpf (3 ss; *n* = 22), 12.5 hpf (6 ss; *n* = 28), 14 hpf (9 ss; *n* = 30), and 15.5 hpf (12 ss; *n* = 26) from top to bottom. Red, blue and green triangles indicate the ALD, FRS, and PLD, respectively. (E) Dependence of times to FRS and PLD on DAPT washout time. Light blue and green box-whisker plots indicate 5, 25, 75, and 95 percentiles for embryonic experimental data (exp.). Dark blue and green box-whisker plots indicate those for simulation data (sim.). The white bars mark the median. The PSM shortening, change in cell advection pattern and increase in the coupling strength are combined in the model, see also *Figure 4—videos 3* and *4*. The lack of information about the formation of final segments in embryos precludes simulations for the latest washout (12 ss), see the text. (F) Spatial distribution of segment boundary defects. Symbols indicate embryonic experimental data and lines indicate simulation data. Grey dashed vertical line across panels is a guide to the eye. In (A-C), (E), results of 100 realizations of simulations with each washout timing are

*Figure 4 continued on next page*

*Figure 4 continued*

plotted. Parameter values for numerical simulations are listed in **Supplementary files 1** and **2**. Source data for (D-F) is available in **Figure 1—source data 1**.

The online version of this article includes the following video and figure supplement(s) for figure 4:

**Figure supplement 1.** PSM shortening decreases time to PLD whereas it does not affect time to FRS.

**Figure supplement 2.** Change in advection pattern increases the time to PLD for earlier DAPT washout.

**Figure supplement 3.** Dependence of embryonic FRS and PLD on DAPT washout time.

**Figure supplement 4.** Decrease in the PSM radius $r$ over developmental stages reduces both time to FRS and PLD.

**Figure supplement 5.** Decrease in time to FRS by an increase in the coupling strength over developmental stages in simulations.

**Figure supplement 6.** Trajectories of phase vortices in the physical model including PSM shortening, changes in advection pattern and coupling strength.

**Figure supplement 7.** ALD, FRS, and PLD calculated with the spatial distribution of defective segments.

**Figure supplement 8.** Dependence of single and double defects on DAPT washout timing.

**Figure 4—video 1.** Resynchronization simulation with the changes in PSM length.

https://elifesciences.org/articles/61358#fig4video1

**Figure 4—video 2.** Resynchronization simulation with the changes in PSM advection pattern.

https://elifesciences.org/articles/61358#fig4video2

**Figure 4—video 3.** Resynchronization simulation with the changes in PSM length, advection pattern and value of coupling strength.

https://elifesciences.org/articles/61358#fig4video3

**Figure 4—video 4.** Resynchronization simulation with the changes in PSM length, advection pattern and value of coupling strength.

https://elifesciences.org/articles/61358#fig4video4

---

time $t_{wash\text{-}out}$, and recorded the developmental time taken from washout to observation of FRS and PLD, termed the time to FRS (FRS – $t_{wash\text{-}out}$) and time to PLD (PLD – $t_{wash\text{-}out}$).

In the absence of tissue shortening or a changing cell advection pattern, the times to FRS and PLD were not affected by washout time, as expected, **Figure 4A**. We analyzed the consequence of PSM shortening on PLD while keeping all the other parameters constant over time, **Figure 4B**, **Figure 4—figure supplement 1** and **Figure 4—video 1**. For simplicity, we assumed that the PSM length decreased linearly over time in the simulation, **Figure 4—figure supplement 1A**. PSM shortening decreased the time to PLD for later washout times, **Figure 4B**, because in a shorter PSM at later somite stages phase vortices reached the anterior end of the PSM more quickly, **Figure 4—figure supplement 1B**. With higher speed of PSM shortening, the time to PLD after washout became shorter, **Figure 4—figure supplement 1C–F**. As expected from the independence of FRS on constant PSM length, PSM shortening did not affect FRS, **Figure 4B** and **Figure 3—figure supplement 5**.

We next analyzed the effect of a change in cell advection pattern on PLD, keeping all the other parameters constant over time, **Figure 4C**, **Figure 4—figure supplement 2** and **Figure 4—video 2**. In the model, we represented the change in the advection pattern in a simplified way such that at earlier somite stages ($t < t_g$) the local strain rate was larger in the anterior region of the PSM, whereas the strain rate became larger in the posterior region at later stages ($t > t_g$), **Figure 4—figure supplement 2A**. We found that such a change in advection pattern increased time to PLD for earlier DAPT washout, **Figure 4C**. If the change in advection pattern occurred at later developmental stages, time to PLD for later washouts was also increased, **Figure 4—figure supplement 2D–G**. As described above, the cell advection pattern in the PSM underlies the transport of phase vortices. When advection did not occur in the posterior PSM at earlier somite-stages, the movement of phase vortices relative to the tailbud was slowed in that region, delaying PLD for earlier washout timing, **Figure 4—figure supplement 2A–C**. As expected from the independence of FRS on constant advection patterns, a change in pattern did not affect FRS, **Figure 4C** and **Figure 4—figure supplement 2D–G**.

Taken together, these results predict that the changes in tissue length and cell advection pattern observed in the embryo over developmental time may have an impact on resynchronization

dynamics, reflecting in the decrease in difference between FRS and PLD. Thus, changing tissue properties may explain the difference between early and late washout phenotypes.

## Embryonic segment recovery in zebrafish depends on timing of DAPT washout

To test the theoretical predictions about the influence of tissue-level changes on the time from washout to FRS and PLD over developmental stages, we next performed DAPT washout experiments, as illustrated in *Figure 4D*, in which DAPT was removed at different times ($t_{wash-out}$) between 9.5 and 15.5 hpf. We visualized the resulting distribution of segment defects along the axis on both left and right-hand sides of the embryo and identified the FRS and PLD, arrowheads in *Figure 4D*.

We found that FRS and PLD both increased with later washout times, *Figure 4—figure supplement 3*. In accordance with the prediction of the model, the time to PLD (PLD – $t_{wash-out}$) decreased gradually over developmental time, *Figure 4E*. In contrast, the experimentally observed gradual decrease in time to FRS (FRS – $t_{wash-out}$) in *Figure 4E* was not expected from the simulations, *Figure 4B,C*. After washout, it took ~13 segments to observe FRS for an earlier washout time, whereas it took eight segments for a later washout time. Embryos yielded more scattered, non-continuous defects with earlier than with later washouts, *Figure 4D* and *Figure 1—figure supplement 1C–G*.

Combined, these results revealed the transition between early and late washout segmentation phenotypes. The observed decrease in the time to PLD was qualitatively consistent with the theoretical predictions. However, the expectation that FRS would be independent of washout timing, based on its insensitivity to global tissue properties of PSM shortening and changing cell advection patterns in the model, was not observed experimentally. We therefore hypothesized that FRS might instead be affected by mechanisms that determine the level of local synchronization.

## Prediction of embryonic FRS from simulated DAPT washout timing and increasing coupling strength

Local synchronization is thought to be driven by local intercellular interactions (*Delaune et al., 2012*; *Horikawa et al., 2006*; *Jiang et al., 2000*; *Riedel-Kruse et al., 2007*). The intensity of such local interactions is described in the theory by the coupling strength $\kappa_0$ between neighboring oscillators. As discussed previously, coupling strength can strongly influence FRS, *Table 1* and *Figure 3—figure supplement 8*. Although FRS can also be influenced by the PSM radius, which becomes smaller with developmental stage, the effect of change in the PSM radius was weaker than the coupling strength within the biologically plausible range, *Figure 3—figure supplement 7* and *Figure 4—figure supplement 4*. Therefore, we tested whether a changing coupling strength could describe the dependence of time to FRS on $t_{wash-out}$ observed in experiments.

For simplicity, we assumed that, in the absence of any perturbation, the coupling strength increased as a linear function of time in the simulation, *Figure 4—figure supplement 5A*. An increase in coupling strength over developmental stages in the embryo could be caused by an increase in the abundance or activity of Delta and Notch proteins in cells (*Wright et al., 2011*; *Liao et al., 2016*; *Haddon et al., 1998*; *Westin and Lardelli, 1997*), an increase in the contact surface area between neighboring cells, or some other mechanism.

A temporal increase in coupling strength in the simulation, allowing it to double by ~15 ss, led to a decrease in the time to FRS that reproduced the experimental results, *Figure 4—figure supplement 5B*. The time to PLD also decreased with $t_{wash-out}$. However, the magnitude of reduction in time to PLD was similar to that in time to FRS, meaning that this effect would not be expected to contribute to the observed experimental reduction in the difference between PLD and FRS. These results indicate that the increase in the coupling strength over somite stages alone is sufficient to realize the dependence of time to FRS on $t_{wash-out}$, whereas the global tissue parameters contribute to the behavior of PLD observed in experiments.

## Embryonic segmentation defect patterns are captured in simulations by PSM shortening, change in cell advection pattern and an increase in coupling strength

Finally, we simulated a resynchronizing PSM with all the three effects described above combined: the changes in PSM length and cell advection pattern over time were imposed by existing experimental data, the change in coupling strength was motivated by the results in the previous section, and all other parameters remained unchanged, *Figure 4E*, *Figure 4—figure supplement 6* and *Figure 4—videos 3* and *4*.

The beginning and the end of somitogenesis are special. The segmentation clock becomes active and rhythmic before somitogenesis starts, during epiboly (*Riedel-Kruse et al., 2007*). At this stage, the embryo undergoes dramatic morphological changes that we do not describe with the current model, which instead describes the PSM shape from 0 ss onwards. For the end of somitogenesis, there is a lack of quantitative information about the formation of the final segments that precludes constraining the theory at this late stage. Therefore, we simulated DAPT washout times from the beginning of somitogenesis 0 ss ($t_{wash-out}$ = 9.5 hpf) until 9 ss ($t_{wash-out}$ = 14 hpf), where the model could be well parametrized and provided a fair description of tissue shape changes.

The requirement for many realizations to compute FRS and PLD precluded application of standard fitting procedures for determining parameter values in the model. Instead, we used parameter values close to those observed in embryos. We found a decrease in time to PLD with the magnitude of the decrease greater than time to FRS, as observed in the experimental data, *Figure 4E*. Inclusion of the PSM shortening and change in cell advection pattern in simulations recapitulated the experimental observation that the difference between PLD and FRS became smaller with later washout time. PSM shortening decreased time to PLD, without affecting FRS, thereby reducing the difference between PLD and FRS for later washout time, *Figure 4B*. The slower cell advection in the posterior PSM at earlier time in simulations delayed PLD without affecting FRS, in principle enlarging the difference between PLD and FRS for earlier washout time, *Figure 4C*.

In summary, a change in the coupling strength was sufficient to reproduce the behavior of FRS. Combined effects of the PSM shortening and cell advection pattern were the dominant factors that generated the behaviors of PLD in simulations. Thus, the physical model quantitatively reproduced the behaviors of FRS and PLD, suggesting that the physiologically plausible changes in these tissue parameters may underlie behaviors observed in the experiment.

## Predicted segment defect distribution confirmed in zebrafish resynchronization assay

We showed that the model could capture the onset of segment boundary recovery and its completion, quantified by FRS and PLD, respectively. However, segment recovery is a complex gradual process reflected in intermingled segment defects. Therefore, we further tested whether the model captured this gradual recovery process between its onset and completion with data that were not used to develop it: the spatial distribution of segment boundary defects along the embryonic body axis.

We used the same parameters listed in *Supplementary file 2* that we established to describe FRS and PLD, *Figure 4E*. Since simulations started from completely random initial phases, the initial fraction of defective segments was one. The fraction of defective segments decreased from one to zero along the body axis after DAPT washout, shifting posteriorly for later $t_{wash-out}$, *Figure 4F*. We then compared the simulated axial distribution of defective segments with embryonic DAPT washout experiments. We restricted the comparison to the washout phase of the experiment. We counted the number of defective segments along the axis in embryos and defined the fraction of defective boundaries at each segment position, *Figure 4F*. The distributions of defective segments were similar between left and right sides of embryos, *Figure 4—figure supplement 7A*. After washout, the fraction of defective segment boundaries gradually decreased, and eventually it became zero, suggesting that synchronization was fully recovered at that time. As DAPT was washed out at increasingly later times, defective segment boundaries continued to more posterior locations, in agreement with simulations, *Figure 4F*.

From this distribution in embryos, we could compute ALD, FRS, and PLD using a probabilistic theory that assumed left-right independence, see Appendix and *Figure 4—figure supplement 7*. This

distribution also explained the ratio of single defects, where either the left or right segment was defective at a segment locus, to double defects, where both left and right segments were defective, Appendix and *Figure 4—figure supplement 8*. This agreement between experimental data and probability theory for the fraction of single defects indicates that recovery occurred independently between left and right PSMs, *Figure 4—figure supplement 8C*. In summary, the physical model predicted the segment defect distribution, providing a thorough description of synchrony recovery.

## Discussion

The segmentation clock produces dynamic patterns that determine the formation of vertebrate segments. This multicellular clock, consisting of thousands of cells that make the PSM and tailbud, produces a kinematic wave pattern. The integrity of this wave pattern relies on local synchronization of the oscillators mediated by Delta-Notch cell-cell signaling. Our current view of synchrony is largely informed by desynchronization experiments in which cells were uncoupled by interfering with Delta-Notch signaling, resulting in a loss of wave patterns. In contrast, resynchronization, where oscillators re-establish coherent rhythms from a desynchronized state, can be used to probe how tissue-scale collective patterns arise from local interactions during morphogenesis.

In this study, we applied a combination of experiments and theory to explore the recovery of normal body segments during the resynchronization of oscillators. Our surprising experimental discovery was regions of normal and defective segments intermingled along the body axis following DAPT washout. Since we seek to capture pattern recovery at multiple scales, our new physical model draws from previous work describing cellular oscillations with an effective phase variable together with local intercellular interactions (*Uriu et al., 2017*; *Morelli et al., 2009*), as well as larger scale mechanics such as cell movements (*Uriu et al., 2017*; *Uriu and Morelli, 2014*) and tissue shape changes (*Jörg et al., 2015*). One of the novelties of this work is to combine these descriptions in a coherent framework for the first time. The second key novelty is the framework itself, which uses tissue geometry linked with changing mechanical and biochemical properties, such as cellular advection profile and coupling strength.

The choice of a phase description for the cellular oscillator is motivated by its generality and by prior success in analyzing coupling in the segmentation clock (*Riedel-Kruse et al., 2007*; *Uriu et al., 2017*; *Herrgen et al., 2010*). Core genetic components of the cellular oscillator have been identified and their dynamics can be described by detailed delay models (*Schröter et al., 2012*; *Lewis, 2003*; *Monk, 2003*; *Horikawa et al., 2006*; *Ay et al., 2013*; *Hirata et al., 2004*; *Jensen et al., 2003*), but since we do not measure any of the components in these networks, the choice of phase oscillators captures the core oscillatory behavior without additional underconstrained parameters (*Kotani et al., 2012*; *Kotani et al., 2020*). Thus, although we do not anticipate any qualitative differences between these modeling approaches, future work could include such a detailed description of oscillatory genes, potentially allowing a more direct connection to mutant conditions or imaging experiments.

The model qualitatively explains the formation of these intermingled defective segments by the emergence of persistent phase vortices in the posterior PSM and their advection through the tissue to the anterior. The intermingled defects span 1 or 2 segments in the embryonic data, but are not multiples of the local segment length. In simulations with the parameter set we determined here, the defects have the same characteristics. The vortices arise from the local coupling of desynchronized oscillators and the advection is a consequence of axis elongation. As vortices arrive at the anterior end, their mis-oriented local phase patterns result in segment defects, before global pattern recovery is achieved. Thus, the intermingled regions are explained by the intermittent arrival of vortices with a size of approximately one segment. Note that vortices only form during resynchronization, being seeded by the random phases of the desynchronized oscillators and requiring local coupling, and not during desynchronization, which starts by the loss of coupling in a population with a locally smooth distribution of phase.

Importantly, for quantitative comparison between simulations and experiments, we needed to introduce observables such as an index of off-lattice vorticity, and a local order parameter to distinguish normal and defective segments. It was also necessary to incorporate global features of the PSM and tailbud that are observed in the embryo, such as changing tissue length, a change in the advection pattern and a gradient of cell mixing. To achieve a description of faster recovery at later

developmental stages, as indicated by shorter time to embryonic FRS, our model includes a time-dependent increase in effective coupling strength. This is plausible from existing data of Delta and Notch gene expression, for example, but remains an expectation of the current work. Nevertheless, simulations of the model confirm that stronger local coupling leads to faster resynchronization and recovery of the gene expression pattern, as expected from previous experimental work (*Liao et al., 2016*). To reduce computational complexity and time, in our model we neglected the coupling time-delays that are thought to occur in the segmentation clock (*Herrgen et al., 2010*; *Oates, 2020*; *Shimojo and Kageyama, 2016*; *Yoshioka-Kobayashi et al., 2020*). We do not anticipate their inclusion would affect our main results, since other systems with time-delayed coupling exhibit vortex formation (*Jeong et al., 2002*), although a quantitative description of some phenotypes may need to account for coupling delays in Delta-Notch signaling.

Because our data was captured from snapshots at a late developmental stage, for simplicity we assumed a constant segment length. However, it has been long recognized that segment length at the time of formation gradually decreases in the posterior of all extending vertebrate embryos (*Gomez et al., 2008*; *Schröter et al., 2008*). These early length differences are gradually reduced and eventually eliminated by subsequent growth in the zebrafish (*Lleras Forero et al., 2018*). It is possible that a quantitative description should also take the phenomenon of changing segment length into account. This would require a timelapse analysis of the forming segments during perturbations and a greater understanding of the mechanism underlying the determination of and change in segment lengths during development (*Ishimatsu et al., 2018*; *Simsek and Özbudak, 2018*), and could be accommodated in the framework of the current model. In summary, the formulation of the model allows a quantitative comparison to experimental data and a phenomenological understanding of pattern recovery.

Vortices could arise in other models of the segmentation clock with similar local interactions (*Uriu et al., 2017*; *Morelli et al., 2009*; *Uriu and Morelli, 2014*; *Hester et al., 2011*). Indeed, vortex formation is a common feature of systems that can be described with locally coupled polar variables, in both excitable and oscillatory media (*Mikhailov and Showalter, 2006*). These include biological systems such as cAMP patterns in aggregating populations of *Dictyostelium* cells (*Durston, 2013*) and spiral patterns in heart tissue (*Winfree, 1980*), planar cell polarity (*Burak and Shraiman, 2009*), chemical systems like the BZ reaction (*Winfree, 1980*; *Kuramoto, 1984*; *Winfree, 1972*) and in physical systems generally (*Kosterlitz, 2016*; *Kosterlitz and Thouless, 1973*). In unconstrained and homogeneous oscillatory media, vortices can grow to system-size length scales and are very stable on their own (*Mikhailov, 1990*). However, the effect on size and stability of concurrent features in our model such as the particular geometrical confinement, as well as the existence of a frequency profile and non-uniform advection has not been examined. Thus, the relationships between vortex size, frequency of arrival, orientation and the underlying processes such as coupling strength and tissue geometry in our model are interesting topics for future exploration.

Recent experiments using cells isolated from mouse tailbuds and reaggregated to form so-called emergent PSM have shown the emergence of striking wave patterns that depend on Notch signaling, suggesting that local interactions can drive large-scale dynamical features (*Hubaud et al., 2017*; *Tsiairis and Aulehla, 2016*). The relationship between such features and normal versus defective segment boundary formation remains to be explored. Since phase

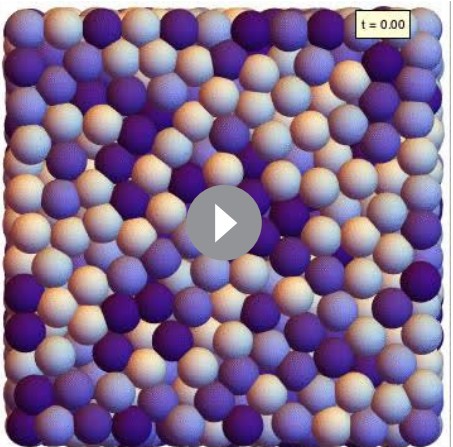

**Video 1.** Formation of phase vortices in a resynchronization simulation in a cuboid domain, $110 \times 110 \times 55\ \mu m^3$. The color indicates $(1 + \sin\theta_i)/2$. The number of oscillators is $N = 998$. Frequencies of all the oscillators are identical, $\omega = 0.2094\ min^{-1}$. Values of the other relevant parameters in *Equation (1)* in the supporting information are: $\kappa_0 = 0.07\ min^{-1}$, $\kappa_s = 0\ min^{-1}$, $D_\theta = 0.0013\ min^{-1}$, $\mu = 8.71\ \mu m\ min^{-1}$, $d_c = 11\ \mu m$, $\mu_b = 20\ \mu m\ min^{-1}$, and $r_b = 1\ \mu m$.
https://elifesciences.org/articles/61358#video1

vortices arise naturally in systems of locally coupled oscillators starting from disordered initial conditions, *Video 1*, we predict that these structures will form also in mammalian PSM tissue culture systems (*Hubaud et al., 2017*; *Tsiairis and Aulehla, 2016*; *Diaz-Cuadros et al., 2020*; *Matsuda et al., 2020*). The framework of our model with different geometries will facilitate analysis of dynamics in these and other collective cellular systems with both local interactions and tissue-level deformations.

In simulations, the kinematics of phase vortices across the PSM is determined by global tissue properties, including the PSM length and its cell advection pattern. In this way, the recovery of a gene expression wave pattern across the PSM and subsequent normal boundary formation is set both by the timescales of local synchronization through intercellular interactions, and also by those of morphological processes. In addition, this result implies that the quantification of global tissue parameters will be necessary to obtain quantitative agreement between theory and experiment. Temporal changes in PSM length have been measured in various species (*Soroldoni et al., 2014*; *Gomez et al., 2008*; *Ishimatsu et al., 2018*), and cell advection patterns across the PSM have been investigated (*Lawton et al., 2013*; *Mongera et al., 2018*; *Steventon et al., 2016*; *Bénazéraf et al., 2010*). Involvement of these global tissue properties discriminates the resynchronization of the segmentation clock from its desynchronization, which is dominated by local cellular properties such as noise in clock gene expression (*Jiang et al., 2000*; *Riedel-Kruse et al., 2007*; *Keskin et al., 2018*; *Jenkins et al., 2015*). Furthermore, morphological tissue development distinguishes the synchronization of the segmentation clock from that of other biological oscillator systems in spatially static tissues, such as the suprachiasmatic nucleus in the mammalian circadian clock (*Webb and Oates, 2016*).

Numerical simulations of the physical model showed that the first recovered segment FRS and the posterior limit of defects PLD contain different information about resynchronization. From FRS, we estimated properties of local synchronization, such as the coupling strength between genetic oscillators. In contrast, PLD is influenced by global tissue properties, such as tissue length and cell advection pattern, that determine the transport of persistent mis-oriented phase patterns. Hence, it is important to choose these measures appropriately depending on the question addressed. For example, recent studies suggested that the level of cell mixing observed in the tailbud promotes synchronization of genetic oscillation (*Uriu et al., 2017*). This effect appears in PLD, but not in FRS, because the elevated mixing in the tailbud prevents formation of the local phase patterns at the posterior PSM. On the other hand, FRS can be a better measure for the coupling strength (*Liao et al., 2016*), because it is less affected by the other tissue parameters than PLD.

Based on the new theoretical framework developed in this study, we have proposed that the distinctive intermingling of well-formed and defective segments seen during recovery arises from persistent phase vortices. Furthermore, we expect the kinetics of phase vortex formation and transport to change throughout development. We have presented quantitative experimental evidence on the size of defects, the shape of the spatial defect distribution and the left-right independence of defects that supports these theoretical predictions. The occurrence of vortices and the role that these transient patterns have in causing intermingled defects remain to be directly observed. Such transient patterns would be difficult to recognize in snapshots of the segmentation clock due to the restricted system geometry, the discrete character of cellular tissue, and the difficulty of determining phase from a single time point, and consequently may have been overlooked in previous experiments. However, they should be within reach of techniques for perturbation of cell coupling combined with live imaging over the long durations required to observe them in the embryo (*Yoshioka-Kobayashi et al., 2020*; *Wan et al., 2019*). Such experiments will be challenging nevertheless, as quantitatively testing the model will require combining (i) imaging, segmentation and tracking of single cells within the embryo across appropriate developmental stages; (ii) an approach to reliably estimate phase from segmentation clock transgenic reporter signals with amplitude fluctuations; (iii) data that allows the calculation of vorticity; and (iv) a method of correlating the passage of vortices with the resulting intermittent segment defects in the same embryo. The use of mutants with reduced coupling strength (*Liao et al., 2016*) may increase the time interval over which vortices are produced, and may thus facilitate their observation.

In conclusion, our study of resynchronization has revealed how the segmentation clock can be influenced by global developmental processes such as tissue length change, cell advection pattern and cell movement, as well as local coupling strength between cells. Our findings suggest that segmental pattern recovery occurs at two scales: local pattern formation and transport of these patterns

through tissue deformation. Other developing systems such as *Dictyostelium* colony aggregation show similar dynamics, with individual motile cells interacting via local signaling to generate spiral waves that guide the formation of large multicellular fruiting bodies (*Durston, 2013*). In the case of the elongation of the *Drosophila* pupal wing, local cell rearrangements within the epithelium combine with tissue-level pulling forces applied by a neighboring tissue to form the final pattern (*Etournay et al., 2015*). The hallmark of these systems is an interplay between locally driven interactions and global morphological changes, pointing to a common principle of pattern dynamics within developing tissues.

## Materials and methods

### Animals and embryos

Zebrafish (*Danio rerio*) adult stocks were kept in 28°C fresh water under a 14:10 hr light:dark photoperiod. Embryos were collected within 30 min following fertilization and incubated in petri dishes with E3 media. Until the desired developmental stages (*Kimmel et al., 1995*), embryos were incubated at 28.5°C. For whole mount in situ hybridization, PTU (1-phenyl 2-thiourea) at a final concentration of 0.003% was added before 12 hr post fertilization (hpf) to prevent melanogenesis. All wildtypes were *AB* strain.

### DAPT treatment and washout

DAPT treatment was carried out as previously described (*Riedel-Kruse et al., 2007*). 50 mM DAPT stock solution (Merck) was prepared in 100% DMSO (Sigma) and stored in a small volume at −20°C. Embryos in their chorions were transferred to 12-well plates at 2 hpf in 1.4 ml E3 medium with 20 embryos per well. 50 μM DAPT in E3 medium was prepared immediately before the treatment. To prevent precipitation, the DAPT stock solution was added into E3 medium while vortexing, and then filtered by 0.22 μm PES syringe filter (Millipore). DAPT treatment was initiated by replacing E3 medium with E3/DAPT medium at desired stage. At 9.5 (0 somite stage: ss), 11 (3 ss), 12.5 (6 ss), 14 (9 ss), or 15.5 hpf (12 ss), DAPT was washed out at least twice with fresh E3 medium + 0.03% PTU. Embryos were dechorionated and fixed at 36 hpf. All experimental steps were incubated at 28.5°C, except for short operations, for example, washing out, which were at room temperature.

### Whole-mount in situ hybridization and segmental defect scoring

In situ hybridization was performed according to previously published protocols (*Thisse and Thisse, 2008*). Digoxigenin-labeled *xirp2a* (clone: *cb1045*) riboprobe was as previously described (*Riedel-Kruse et al., 2007*). Stained embryos were visually scored under an Olympus SZX-12 stereomicroscope and images were acquired with a QImaging Micropublisher 5.0 RTV camera. Defective segment boundaries were scored as previously described (*Riedel-Kruse et al., 2007*), with the addition that left and right (LR) sides of the embryo were scored and assessed separately. Since boundary formation in unperturbed embryos is extremely reliable, with errors occurring in less than 1 in 1000 embryos, any interruption or fragmentation to the boundary, and/or alterations in spacing, or alignment was recorded as a defect. The following observables were collected for each LR side: an anterior limit of defects (ALD), that is, the position of the first defective boundary; a posterior limit of defects (PLD), that is, the position of the last defective boundary; and the first recovered segment (FRS), that is, the position of the first normal segment after the segment 9, as described below.

Each segment has anterior and posterior boundaries. For the segment $i$ ($i$ = 1, 2, 3,. .,), the anterior and posterior boundaries were numbered as $i − 1$ and $i$, respectively, *Figure 1—figure supplement 1A*. Both ALD and PLD were numbered using the posterior boundary of the defective segment. For example, if the $j$th segment boundary was the last defective boundary, PLD was numbered as $j$. FRS was numbered using the anterior boundary of the recovered segment. For example, if the $k$th segment boundary was the first normal boundary after washout, the first normal segment was segment $k$, and FRS was numbered as $k − 1$, *Figure 1—figure supplement 1A*. With this definition of FRS, if the first normal segment boundary after the washout was located immediately after the last defective boundary, the difference between PLD and FRS, PLD − FRS, was 0, *Figure 1—figure supplement 1C–G*. Definitions of FRS and PLD for simulation date were the same as those for experimental data written here, and described in a later section.

## Physical model of PSM and tailbud cells

### 3D tissue geometry

We model the PSM and tailbud as a U-shaped domain in a 3D space, *Figure 2—figure supplement 1A*. The left and right PSMs are represented as two tubular domains $\Omega_l$ and $\Omega_r$, respectively with radius $r$. The tailbud is described as a half toroidal domain $\Omega_t$ with a larger radius $R$ and smaller radius $r$.

We implement this U-shaped domain in the spatial coordinate system as follows. The x-axis is along the anterior-posterior axis of the PSM. We set the initial position of the anterior end of the PSM at $x = 0$ and the posterior tip of the tailbud at $x = L_x$. We denote the position of the anterior end of the PSM at time $t$ as $x_a(t)$, so $x_a(0) = 0$. The length of the tissue is $L = L_x - x_a$. $\mathbf{X}_c$ denotes the position of the center of torus core curve. The position of posterior tip of the tailbud can then be written as $L_x = X_c + R + r$. The y-axis points along the left-right axis and z-axis along the dorsal-ventral axis of an embryo.

### Reference frames: Lab reference frame

To describe cell movements and tissue deformations it is important to define a reference frame. A natural choice may be a reference frame which is at rest in the Lab, termed a Lab reference frame. For instance, the origin of x-axis can be set at the initial position of the anterior end of the PSM at $t = 0$. We write the position of the posterior tip of the tailbud in this Lab reference frame as $x_t^{(L)}(t)$, where superscript $(L)$ indicates variables in the Lab reference frame. Because an embryo elongates posteriorly, $dx_t^{(L)}(t)/dt > 0$. The position of anterior end of the PSM in the Lab reference frame $x_a^{(L)}(t)$ also changes over time due to the formation of new segments. In this reference frame, PSM length is $L(t) = x_t^{(L)}(t) - x_a^{(L)}(t)$. The rate of change of PSM length is $dL(t)/dt = dx_t^{(L)}(t)/dt - dx_a^{(L)}(t)/dt$. PSM length is constant if the velocity of the tailbud is the same as that of the anterior end of the PSM, that is $dx_t^{(L)}(t)/dt = dx_a^{(L)}(t)/dt$, but it will be changing over time whenever $dx_t^{(L)}(t)/dt \neq dx_a^{(L)}(t)/dt$. In this Lab reference frame, the position of a cell $x_c^{(L)}(t)$ may change both due to tissue elongation and due to tissue deformations such as local tissue stretch.

### Reference frames: tailbud reference frame

In this study however, we mostly use the tailbud reference frame $(t)$ unless noted otherwise. In this reference frame we measure the position of cells and tissue landmarks from the tailbud. The position of the tailbud is fixed at $x_t^{(t)}(t) = 0$ for all time $t$, so $dx_t^{(t)}(t)/dt = 0$. The position of a tissue landmark $l$ is $x_l^{(t)}(t) = x_l^{(L)}(t) - x_t^{(L)}(t)$. In this tailbud reference frame, the velocity of a landmark is zero if it moves with the same velocity as the tailbud in the Lab frame. In other words, a tissue landmark moves relative to the tailbud when their velocities in the Lab reference frame are different. The positions of cells in the tailbud reference frame may change over time, and this will enter as cell advection in the cell equations of motion as we describe below. In the following, we omit the superscript $(t)$ for the tailbud reference frame to alleviate the notation.

### Reference frames: computational implementation

For the implementation of the tailbud frame for numerical simulations, we fix the position of the posterior tip of the tailbud at $x = L_x$. In contrast, cells positions and other tissue landmarks such as the anterior end of the PSM change over time. We model cell displacement relative to the tailbud as cell advection in the anterior direction as explained below.

### Cell mechanics and equation of motion

We model PSM and tailbud cells as particles in the 3D domain. We denote the number of cells in the PSM and tailbud at time $t$ as $N(t)$. State variables for these cells are their position $\mathbf{x}$ in the domain, the phase $\theta$ of their genetic oscillators and the unit vector $\mathbf{n}$ representing cell polarity for intrinsic cell movement. Position of cell $i$ at time $t$ is denoted as $\mathbf{x}_i(t) = (x_i(t), y_i(t), z_i(t))$ $(i = 1, 2, 3, ..., N(t))$. We describe cellular motion by the following over-damped equation based on the previous study (*Uriu et al., 2017*):

$$\frac{d\mathbf{x}_i}{dt} = \mathbf{v}_d(\mathbf{x}_i) + v_0(\mathbf{x}_i)\mathbf{n}_i(t) + \sum_{j=1, j \neq i}^{N} \mathbf{F}(\mathbf{x}_i, \mathbf{x}_j) + \mathbf{F}_b(\mathbf{x}_i), \tag{1}$$

where $\mathbf{v}_d(\mathbf{x}_i)$ is cell advection from posterior to anterior, $v_0(\mathbf{x}_i)$ represents the speed of intrinsic cell movement, $\mathbf{n}_i(t)$ is the cell polarity pointing the direction of intrinsic movement, $\mathbf{F}(\mathbf{x}_i, \mathbf{x}_j)$ is a physical contact force between cells $i$ and $j$, and $\mathbf{F}_b(\mathbf{x}_i)$ is a boundary force that confines cells within the U-shaped domain. We explain each of these terms below.

## Elongation, strain rate, and cell advection

To introduce tissue elongation and deformations we first discuss the movement of tissue landmarks and cells in the Lab reference frame. Then, we switch to the tailbud reference frame where the tissue elongation results in cell advection from the posterior to the anterior directions.

In the Lab reference frame, segments and cells within do not move. The tailbud, together with cells at that point, moves away from segments with a velocity $\mathbf{v}_t^{(\mathrm{L})}$. We call *elongation* to this out-growth of the tissue. As cells differentiate into a segment, the anterior end of the PSM also moves after the tailbud, with a velocity $\mathbf{v}_a^{(\mathrm{L})}$. If this velocity matches that of the tailbud $\mathbf{v}_a^{(\mathrm{L})} = \mathbf{v}_t^{(\mathrm{L})}$, the length of the PSM remains constant. If this velocity differs from that of the tailbud $\mathbf{v}_a^{(\mathrm{L})} \neq \mathbf{v}_t^{(\mathrm{L})}$, the length of the PSM changes over time, causing a *global tissue deformation*. At these PSM ends we have boundary conditions for the cells velocities: (i) a cell at the tailbud moves with the same velocity as the tailbud and (ii) a cell at the anterior end of the PSM is at rest, like neighboring cells within a segment, even though the anterior end of the PSM itself is moving in the Lab frame. Within the PSM, cells are subject to an advection velocity field $\mathbf{v}^{(\mathrm{L})}(\mathbf{x})$ that satisfies these boundary conditions. This velocity field producing *internal local deformations* is caused by internal strains described below. The shape of this velocity field determines the kind of local deformations. For example, a linear velocity field produces uniform deformations, while a non-linear velocity field produces non-uniform deformations, as the piecewise linear function described below in the implementation.

To see the relation between the velocity field and underlying internal strains, let $x_1(t)$ and $x_2(t)$ be the $x$ positions of cells 1 and 2 at time $t$ in the $(\mathrm{L})$ reference frame, where we drop the $(\mathrm{L})$ super-script for notational convenience. We assume that their positions are close to each other, so the distance between them $\Delta x(t) = x_2(t) - x_1(t)$ is small. Due to the presence of an advective velocity field, the velocities of neighboring cells may be different, that is $v_1 \neq v_2$, with $v_1 = dx_1/dt = v(x_1(t))$ and $v_2 = dx_2/dt = v(x_2(t)) = v(x_1(t) + \Delta x(t))$. Thus, during a short time interval $\Delta t$ these cells change their relative position due to this velocity difference, $\Delta x(t + \Delta t) = x_2(t + \Delta t) - x_1(t + \Delta t) \approx \Delta x(t) + (v_2 - v_1)\Delta t$. Thus, there is a local strain $(\Delta x(t + \Delta t) - \Delta x(t))/\Delta x(t) \approx (\partial v/\partial x)\Delta t$, where the velocity field gradient $(\partial v/\partial x)$ is a local strain rate. Thus, the advection velocity field effectively describes the presence of local strains and determines the local tissue deformation. Although such local deformations make cells move apart from each other along the x-axis, intercellular and boundary forces in *Equation (1)* constrain cell distances and we do not observe large density fluctuations due to the velocity field gradient, as can be seen in simulations. Furthermore, where local cell density fluctuations do happen, cell addition described below will bring back the density to its average value.

## Cell advection patterns relative to the tailbud reference frame

Back to the tailbud reference frame, the velocity field is $\mathbf{v}^{(t)}(\mathbf{x}) = \mathbf{v}^{(\mathrm{L})}(\mathbf{x}) - \mathbf{v}_t^{(\mathrm{L})}$, which we refer to as the cell advection pattern $\mathbf{v}_d(\mathbf{x})$ in *Equation (1)* and in the main text. We model different advection patterns of the PSM effectively by changing the spatial derivative of the advection speed in subdomains of the tissue. The advection pattern of the PSM may change at a certain developmental stage as we discussed in the main text. Below, we model the spatial profile of cell advection speed and its temporal change. For simplicity, we divide the PSM into two subdomains, namely the anterior PSM $((x - x_a)/L < x_q)$ and the posterior PSM $((x - x_a)/L > x_q)$, and consider different strain rates $\partial \mathbf{v}_d/\partial x$ for each domain, *Figure 2C*. The advection field $\mathbf{v}_d(\mathbf{x})$ in *Equation (1)* depends on spatial position along the anterior-posterior axis $x$ ($x_a \leq x \leq L$) and is described as (*Figure 2C*):

$$\mathbf{v}_d(\mathbf{x}) = (-v_d(\overline{x}/L), 0, 0)^{\mathrm{T}}, \tag{2a}$$

$$v_d(\chi) = \begin{cases} -\dfrac{v_a - v_p\left(1 - x_q\right)}{x_q}\chi + v_a & \chi \leq x_q, \\ -v_p\chi + v_p & \chi > x_q, \end{cases} \tag{2b}$$

where $\overline{x} = x - x_a$, $v_a > 0$ and $v_p > 0$ are the parameters that determine the advection speed, the superscript $\mathrm{T}$ denotes transposition of the vector, and $L$ is the length of the PSM $L = L_x - x_a$. The slope changes at the position $x_q (0 < x_q < 1)$. Thus, $x_q$ divides the PSM into anterior and posterior subdomains. Within each domain, the strain rate is uniform, whereas it may be different between these two domains. The strain rate, given by the magnitude of the spatial gradient of advection speed, is $v_p$ in the posterior PSM domain and $\left(v_a - v_p\left(1 - x_q\right)\right)/x_q$ in the anterior PSM domain.

## Temporal change in advection pattern

The advection pattern of the PSM in embryos may change at a certain developmental stage. To model the change in the advection pattern of the PSM, we change the value of $v_p$ in *Equation (2)* at time $t = t_g$. We assume that for $t_g$, advection occurs mostly at the anterior part, $v_p < v_a$. For $t_g$, we assume that advection occurs at the posterior part of the tissue $v_p > v_a$.

## Gradient of intrinsic cell movement speed

A cell mixing gradient is observed along the anterior-posterior axis of the PSM (*Lawton et al., 2013*; *Mongera et al., 2018*; *Mara et al., 2007*; *Uriu et al., 2017*). The degree of cell mixing is higher in the tailbud and posterior region of the PSM than anterior region. To model the cell mixing gradient, we assume that the speed of intrinsic cell movement $v_0(\mathbf{x})$ in *Equation (1)* depends on the spatial position $\overline{x} = x - x_a$ along the anterior-posterior axis (*Figure 2B*):

$$v_0(\mathbf{x}) = v_s\left[1 + \left(\frac{1 - \overline{x}/L}{X_v}\right)^h\right]^{-1}, \tag{3}$$

where $v_s$ is the maximum speed at the posterior tip of the tailbud, $X_v$ is the lengthscale of the mobility gradient, and the coefficient $h$ determines the steepness of the mobility gradient, *Figure 3—figure supplement 4A*.

## Cell polarity

The unit vector $\mathbf{n}_i(t)$ in *Equation (1)* represents the polarity of cell $i$ and determines the direction of intrinsic cell movement. In spherical coordinates, $\mathbf{n}_i(t) = (\sin\phi_i(t)\cos\varphi_i(t), \sin\phi_i(t)\sin\varphi_i(t), \cos\phi_i(t))^{\mathrm{T}}$ with $0 \leq \phi_i(t) \leq \pi$ and $0 \leq \varphi_i(t) < 2\pi$. We assume random change of cell polarity by letting the polarity angles $\phi_i$ and $\varphi_i$ perform a random walk. The time evolution of $\mathbf{n}_i$ is described by Langevin equations for these two polarity angles (*Uriu et al., 2017*):

$$\frac{d\phi_i(t)}{dt} = \frac{D_\phi}{\tan\phi_i(t)} + \sqrt{2D_\phi}\xi_{\phi i}(t), \tag{4a}$$

$$\frac{d\varphi_i(t)}{dt} = \frac{\sqrt{2D_\phi}\xi_{\varphi i}(t)}{\sin\phi_i(t)}, \tag{4b}$$

where $D_\phi$ is the polarity noise intensity. White Gaussian noise $\xi_{\phi i}(t)$ and $\xi_{\varphi i}(t)$ satisfy $\langle\xi_{\phi i}(t)\rangle = 0$, $\langle\xi_{\phi i}(t)\xi_{\phi j}(t')\rangle = \delta_{ij}\delta(t - t')$, $\langle\xi_{\varphi i}(t)\rangle = 0$, $\langle\xi_{\varphi i}(t)\xi_{\varphi j}(t')\rangle = \delta_{ij}\delta(t - t')$ and $\langle\xi_{\phi i}(t)\xi_{\varphi j}(t')\rangle = 0$.

To obtain *Equations (4a) and (4b)*, we first consider isotropic diffusion in a plane that is locally tangent to the sphere. Next, we normalize the polarity vector to keep its unit length. Finally, we transform to global spherical coordinates. Formally, we first update the vector $\mathbf{n}_i(t)$:

$$\tilde{\mathbf{n}}_i(t + dt) = \mathbf{n}_i(t) + \sqrt{2D}\xi_{\varphi i}(t)\mathbf{m}_x dt + \sqrt{2D}\xi_{\phi i}(t)\mathbf{m}_y dt, \tag{4c}$$

where $\mathbf{m}_x = (\mathbf{n}_i(t) \times \mathbf{e}_z)/|\mathbf{n}_i(t) \times \mathbf{e}_z|$, $\mathbf{m}_y = (\mathbf{m}_x \times \mathbf{n}_i(t))/|\mathbf{m}_x \times \mathbf{n}_i(t)|$ and $\mathbf{e}_z = (0, 0, 1)$. These unit vectors

$\mathbf{m}_x$ and $\mathbf{m}_y$ define a plane tangent to the unit sphere at position $\mathbf{n}_i(t)$. Random displacement on this tangent plane will move the polarity vector away from the surface of the sphere, so its length $\left|\tilde{\mathbf{n}}_i(t+dt)\right|$ will be larger than 1,

$$\left|\tilde{\mathbf{n}}_i(t+dt)\right| = 1 + 2Ddt + O(dt^2). \tag{4d}$$

We normalize the polarity vector to correct for this radial displacement

$$\mathbf{n}_i(t+dt) = \frac{\tilde{\mathbf{n}}_i(t+dt)}{\left|\tilde{\mathbf{n}}_i(t+dt)\right|}, \tag{4e}$$

and substitute *Equations (4c) and (4d)* into *Equation (4e)* to obtain the differential equation for the unit vector $\mathbf{n}_i(t)$

$$\frac{d\mathbf{n}_i(t)}{dt} = \sqrt{2D}\xi_{\varphi i}(t)\mathbf{m}_x + \sqrt{2D}\xi_{\phi i}(t)\mathbf{m}_y - 2D\mathbf{n}_i(t).$$

Finally, we transform variables from cartesian to spherical coordinates using Ito calculus, and obtain *Equations (4a) and (4b)*. With these Langevin equations, the polarity vector $\mathbf{n}_i$ performs an isotropic and uniform random walk on the surface of a unit sphere.

## Intercellular force

For simplicity, we consider a linear elastic force $\mathbf{F}(\mathbf{x}_i, \mathbf{x}_j)$ to model volume exclusion of cells in *Equation (1)*. If the distance between two cells becomes shorter than a threshold $d_c$ which we term cell diameter, they repel each other (*Uriu et al., 2017*; *Uriu and Morelli, 2014*):

$$\mathbf{F}(\mathbf{x}_i, \mathbf{x}_j) = F(\mathbf{x}_i, \mathbf{x}_j)\frac{\mathbf{x}_j - \mathbf{x}_i}{\left|\mathbf{x}_j - \mathbf{x}_i\right|}, \tag{5a}$$

where $F(\mathbf{x}_i, \mathbf{x}_j)$ is the modulus of intercellular force

$$F(\mathbf{x}_i, \mathbf{x}_j) = \begin{cases} \mu\left(\left|\mathbf{x}_i - \mathbf{x}_j\right|/d_c - 1\right) & \left|\mathbf{x}_i - \mathbf{x}_j\right|/d_c \leq 1 \\ 0 & \left|\mathbf{x}_i - \mathbf{x}_j\right|/d_c > 1, \end{cases} \tag{5b}$$

where $\mu > 0$ is the coefficient for intercellular force.

## Boundary force

$\mathbf{F}_b(\mathbf{x}_i)$ in *Equation (1)* is a confinement force that a cell receives from the boundaries of the domains. Below, we specify this confinement force depending on which tissue domains cells are located in, *Figure 2—figure supplement 1B*.

### In the PSM tubes

In the PSM regions $\Omega_l$ and $\Omega_r$, we consider boundary force in $y$ and $z$ directions. The position of cell $i$ in the right PSM region $\Omega_r$ can be written as

$$\mathbf{x}_i = \left(x_i,\ r + 2R + r_i\cos q_i^{(c)},\ Z_c + r_i\sin q_i^{(c)}\right), \tag{6a}$$

see *Figure 2—figure supplement 1B*. If cell $i$ is in the left PSM region $\Omega_l$, its position can be described as

$$\mathbf{x}_i = \left(x_i,\ r + r_i\cos q_i^{(c)},\ Z_c + r_i\sin q_i^{(c)}\right). \tag{6b}$$

Then, we define frictionless boundary force in the columns as

$$\mathbf{F}_b = \begin{pmatrix} F_{bx} \\ F_{by} \\ F_{bz} \end{pmatrix} = \begin{pmatrix} 0 \\ -\mu_b e^{-\frac{\delta y^{(c)}}{r_b}} \cos q_i^{(c)} \\ -\mu_b e^{-\frac{\delta z^{(c)}}{r_b}} \sin q_i^{(c)} \end{pmatrix}, \tag{7}$$

where $\delta y^{(c)} = \left| (r - r_i)\cos q_i^{(c)} \right|$ and $\delta z^{(c)} = \left| (r - r_i)\sin q_i^{(c)} \right|$.

## In the tailbud half-toroid

Position of cell $i$ within the half-toroidal domain $\Omega_t$ can be expressed as:

$$\mathbf{x}_i = \mathbf{X}_c + \Delta \mathbf{x}_i = \begin{pmatrix} X_c + \Delta x_i \\ Y_c + \Delta y_i \\ Z_c + \Delta z_i \end{pmatrix} = \begin{pmatrix} X_c + R\cos p_i + r_i \cos q_i \cos p_i \\ Y_c + R\sin p_i + r_i \cos q_i \sin p_i \\ Z_c + r_i \sin q_i \end{pmatrix}, \tag{8}$$

where $p_i = \tan^{-1}(\Delta y_i / \Delta x_i)$, $q_i = \tan^{-1}(\Delta z_i \cos p_i / (\Delta x_i - R\cos p_i))$ and $r_i = \Delta z_i / \sin q_i$ (*Figure 2—figure supplement 1B*). We then define the boundary force in the half-toroidal domain as:

$$\mathbf{F}_b = \begin{pmatrix} F_{bx} \\ F_{by} \\ F_{bz} \end{pmatrix} = \begin{pmatrix} -\mu_b e^{-\frac{\delta x}{r_b}} \cos p_i \cos q_i \\ -\mu_b e^{-\frac{\delta y}{r_b}} \sin p_i \cos q_i \\ -\mu_b e^{-\frac{\delta z}{r_b}} \sin q_i \end{pmatrix}, \tag{9}$$

where $\mu_b$ is the coefficient and $r_b$ is the length scale of the boundary force, $\delta x = |R\cos p_i + r\cos p_i \cos q_i - \Delta x_i|$, $\delta y = |R\sin p_i + r\sin p_i \cos q_i - \Delta y_i|$ and $\delta z = |r\sin q_i - \Delta z_i|$.

## Phase equation

The phase dynamics of genetic oscillators in single PSM cells is described by a phase oscillator model (*Riedel-Kruse et al., 2007*; *Uriu et al., 2017*; *Morelli et al., 2009*; *Kuramoto, 1984*):

$$\frac{d\theta_i(t)}{dt} = \omega(\mathbf{x}_i) + \frac{\kappa(t)}{n_{\theta i}(t)} \sum_{|\mathbf{x}_j - \mathbf{x}_i| \leq d_c} \sin\left[\theta_j(t) - \theta_i(t)\right] + \sqrt{2D_\theta}\,\xi_{\theta i}(t), \tag{10}$$

where $\theta_i$ is the phase of cell $i$, $\omega(\mathbf{x}_i)$ is the autonomous frequency at position $\mathbf{x}_i$, $\kappa$ is the coupling strength, $n_{\theta i}$ is the number of neighboring cells interacting with cell $i$ and $D_\theta$ is the phase noise intensity. Interactions occur between touching cells $|\mathbf{x}_j - \mathbf{x}_i| \leq d_c$. The coupling strength $\kappa$ may be time-dependent as described below. $\xi_{\theta i}(t)$ is a white Gaussian noise satisfying $\langle \xi_{\theta i}(t) \rangle = 0$ and $\langle \xi_{\theta i}(t)\xi_{\theta j}(t') \rangle = \delta_{ij}\delta(t - t')$.

We assume the frequency profile $\omega(\mathbf{x})$ along the anterior-posterior axis of the PSM to generate traveling phase waves (*Jörg et al., 2015*; *Morelli et al., 2009*; *Ares et al., 2012*). It is described as $\omega(\mathbf{x}) = \omega_0 U(\mathbf{x})$ where $\omega_0$ is the frequency at the posterior tip of the tailbud. For simplicity, we scale the frequency profile with the PSM length $L$. The function $U(\mathbf{x})$ reads, (*Figure 2D*; *Jörg et al., 2015*):

$$U(\mathbf{x}) = \sigma + (1 - \sigma)\frac{1 - e^{-k\bar{x}/L}}{1 - e^{-k}}, \tag{11}$$

where $\bar{x} = x - x_a$, $\sigma$ denotes the difference in the frequency between anterior and posterior ends of the tissue, and $k$ determines the shape of the frequency profile, *Figure 2D* and *Figure 3—figure supplement 10*.

In *Equation (10)*, we introduced some simplifications. For example, time delays in intercellular interactions with Delta-Notch signaling play important roles in setting the period of collective rhythms and synchronization (*Herrgen et al., 2010*; *Yoshioka-Kobayashi et al., 2020*). However, *Equation (10)* does not include it to reduce computational time. In addition, a recent study suggested the presence of a spatial gradient of noise intensity along the PSM (*Keskin et al., 2018*), but we assume that $D_\theta$ is constant across the tissue. Wildtype embryos with Delta-Notch signaling do not spontaneously form defective segments, suggesting that coupling strength is sufficiently strong

to overcome phase noise. Therefore, we assume that the phase noise intensity is sufficiently lower than coupling strength and approximate its spatial gradient by the zeroth order term.

The phase of cells anterior to the PSM $x<x_a$ is arrested (i.e. $d\theta_i/dt = 0$ for $x_i<x_a$). Then, we obtain advecting stripes for normal traveling waves in the PSM, *Figure 2E*. We consider that the stripes represent segment boundaries and the interval between two consecutive stripes represents segment length as described below.

## DAPT washout at different time points

In this study, we allow the coupling strength in *Equation (10)* to be a time-dependent function $\kappa(t)$. The value of the coupling strength is varied in the presence or absence of DAPT in embryos. Besides the influence of DAPT, the coupling strength in embryonic cells may change intrinsically with developmental stages due to, for example, gradual changes in Delta and Notch protein levels on cell membrane, and/or changes in contact surface areas between neighboring cells. To model such changes in the coupling strength, we assume the following time dependence:

$$\kappa(t) = \begin{cases} 0 & t<t_{wash-out} \\ \kappa_s t + \kappa_0 & t \geq t_{wash-out}, \end{cases} \tag{12}$$

where $t_{wash-out}$ represents the time at which DAPT is washed out in simulations. In the presence of DAPT $t < t_{wash-out}$, there is no interaction between cells. After DAPT washout at $t = t_{wash-out}$, cells restore coupling immediately and interact with each other at finite rates. For simplicity, we assume that the coupling strength changes linearly with time in *Equation (12)* to consider its intrinsic dependence on developmental stages, *Figure 4—figure supplement 5A*. Setting $\kappa_s = 0$ in *Equation (12)* describes a constant coupling strength $\kappa_0$ after DAPT washout. We first analyze resynchronization dynamics with the fixed value of the coupling strength $\kappa_0$ by setting $\kappa_s = 0$, *Figures 3* and *4*, *Figure 3—figure supplements 4–11* and *Figure 4—figure supplements 1*, *2* and *4*. Then, we consider a positive value of $\kappa_s>0$ and let the value of the coupling strength gradually increase to study its effect on time to FRS, *Figure 4E* and *Figure 4—figure supplements 3*, *5*, *6* and *8*.

## PSM shortening

When we model PSM shortening, we consider that the position of the anterior end of the PSM $x_a(t)$ changes over time. For simplicity, we assume that the anterior end of the PSM moves in the posterior direction ($x>0$) at a constant velocity $u_a(u_a>0)$, *Figure 4—figure supplement 1A*:

$$x_a(t) = u_a t. \tag{13}$$

Hence, the length of the PSM becomes shorter as $L(t) = L_x - u_a t$. The speed of the anterior end of the PSM may influence the segment length. For better comparison, we fix the segment length $S$ constant for different values of $u_a$ in simulations. For this, we impose the following condition:

$$v_a + u_a = c, \tag{14}$$

where $v_a$ is the cell advection speed at the anterior end used in *Equation (2)* and $c$ is a constant. Hence, $v_a$ can be expressed as $v_a = c - u_a$. With *Equation (14)*, the segment length $S$ reads $S = c \times T_a$ where $T_a$ is the period of oscillation at the position $x_a$. This anterior period of oscillation $T_a$ is the period that one would measure by recording the phase at the tissue boundary at the anterior end as waves – and cells – pass across this boundary, not to be confused with the period of a single cell there. With this definition, this anterior period $T_a$ matches the segmentation rate. When the PSM length does not change during simulation ($u_a = 0$), the anterior period $T_a$ is equal to the period at the tailbud, $T_a = 2\pi/\omega_0$. When the PSM becomes shorter over time, there is a Doppler effect because waves are traveling toward an approaching anterior boundary (*Soroldoni et al., 2014*; *Jörg et al., 2015*). This Doppler effect changes the readout of anterior period $T_a$, which becomes shorter. To compensate for this effect here, we set $T_a = 30$ min by tuning $\omega_0$ in *Equation (10)* if we include PSM shortening.

## Cell influx and outflux

Due to cell advection, cells reach the anterior end of the PSM $x = x_a$. These cells exit from the domains $\Omega_l$ and $\Omega_r$ at the advection speed $|\mathbf{v}_d(x_a)| = v_a$. We make the simplifying assumption of constant cell density $\varrho \approx \varrho_0$. We implement this assumption in simulations by local density dependent cell addition to the PSM and tailbud, as described below:

1. We measure cell density of the left and right PSM $\varrho_l$ and $\varrho_r$ ($x_a(t) < x < X_c$), respectively, and that of the tailbud $\varrho_t$ ($X_c < x$). Then, we compute the difference between the measured density and the target density $\varrho_0$, $\delta_l = \varrho_l - \varrho_0$, $\delta_r = \varrho_r - \varrho_0$ and $\delta_t = \varrho_t - \varrho_0$.
2. If all $\delta_l$, $\delta_r$, and $\delta_t$ are positive, we do not add a cell to the PSM and tailbud when a cell leaves the PSM from its anterior end $x_a(t)$ due to advection.
3. In contrast, if some of $\delta_l$, $\delta_r$, and $\delta_t$ are negative, we add one cell to the region of which density is smallest. For example, if $\delta_l$ is negative and the smallest, we add a cell to the left PSM. The phase $\theta$ of the added cell is assigned randomly from the uniform distribution between 0 and $2\pi$. The position of the added cell is randomly assigned within the chosen domain. In the anterior end of the embryonic PSM, cells divide less frequently (unpublished observations). For this reason, we do not add cells in the region $x \leq x_a + \zeta$ for the left and right PSM. In this study, we set $\zeta = 100$ μm. We note that adding cells with random phases in this region would be detrimental for the segmented pattern, given that phase disturbances would not have time to synchronize to their neighbors. The cell polarity angles of the added cell are assigned randomly from the uniform distribution between 0 and $2\pi$ for $\varphi$, and between 0 and $\pi$ for $\phi$. The autonomous frequency, speed of intrinsic cell movement and advection speed for the added cell are determined depending on its added position.

## Change in the PSM radius

We examine how change in the PSM radius over time influences resynchronization, *Figure 4—figure supplement 4*. For simplicity, we assume that the PSM radius $r$ decreases uniformly in the U-shaped domain and linearly over time:

$$r(t) = -s_r t + r_0, \tag{15}$$

where $s_r$ ($s_r > 0$) is the magnitude of change in the PSM radius. Because we fix cell density as described in the subsection '*cell influx and outflux*', the number of cells in the U-shaped domain decreases as the PSM radius becomes narrower. Note that although the thinning of PSM in one direction may cause a length extension in other directions to preserve a volume, we do not consider such extension for simplicity.

To simplify the implementation of the model, we separately update cell positions $\mathbf{x}_i$ and the radius size $r(t)$. When updating $r(t)$ into $r(t + \Delta t)$ by *Equation (15)* above, some cells may be left outside the U-shaped domain, $r_i(t + \Delta t) > r(t + \Delta t)$ where $r_i(t + \Delta t)$ is the position of cell $i$ in the radial direction defined in *Equation (6)* or (*Equation 8*) at time $t + \Delta t$. For such cells, we correct their positions after the update of PSM radius as: $r_i(t + \Delta t) \rightarrow r(t + \Delta t) - 2r_b$ where $r_b$ is the lengthscale of boundary force used in *Equations (7) and (9)*.

## Initial condition of simulations

We set the initial position of the anterior end of the PSM $x_a(0)$ as $x_a(0) = 0$. We choose the cell number $N(0)$ to satisfy the cell density $\varrho = \varrho_0$ with a given tissue geometry. Initial positions of cells $\mathbf{x}_i(0) (i = 1, 2, \ldots, N)$ are within the PSM and tailbud domains. To set initial positions of cells, we first randomly locate cells within the U-shaped domain. Then, local stresses caused by the random cell positioning are relaxed by simulating *Equation (1)* without the advection term for 10 min with the integration time step of 0.01 min. In this relaxation process, we include a boundary force at $x = x_a = 0$ to constrain cells within the U-shaped domain. For simulations of embryos without DAPT treatment (i.e. synchronized initial condition), we use a synchronized initial condition $\theta_i(0) = 3\pi/2$ for $i = 1, \ldots, N$. For resynchronization simulations, the initial phase value of cell $i$, $\theta_i(0)$, is chosen randomly from a uniform distribution between 0 and $2\pi$. The values of initial cell polarity angles in *Equation (4)* are also chosen randomly from the uniform distributions between 0 and $\pi$ for $\phi_i(0)$, and between 0 and $2\pi$ for $\varphi_i(0)$. The autonomous frequency, speed of intrinsic cell movement and advection speed are determined by the initial position $\mathbf{x}_i(0)$.

## Parameter values

The values of parameters used in this study are listed in **Supplementary files 1** and **2**. Cell density and diameter of cells are based on the estimation by **Uriu et al., 2017**. The values of parameters that determine intrinsic cell movement and physical forces between cells are set to reproduce experimental data, **Figure 3—figure supplement 4B** (also refer to **Uriu et al., 2017** for details). We use the PSM length within the range reported in **Soroldoni et al., 2014**. The parameters for the frequency profile in **Equation (10)** are based on **Soroldoni et al., 2014**; **Jörg et al., 2015**. Also, the value of the coupling strength $\kappa$ in **Equation (10)** is in the range estimated by **Riedel-Kruse et al., 2007**; **Herrgen et al., 2010**.

Previous studies estimated noise intensity in the segmentation clock employing different theoretical formalisms (**Riedel-Kruse et al., 2007**; **Keskin et al., 2018**; **Jenkins et al., 2015**). The present physical model has two noise sources for phase of oscillation. One is the white Gaussian noise with intensity $D_\theta$ in the phase **Equation (10)**. The other is cell addition with a random phase value described in the section 'Cell influx and outflux'. In desynchronization simulations with $\kappa(t) = 0$ in **Equation (10)**, noise by cell addition alone ($D_\theta = 0$) ruins kinematic phase waves and stripe patterns, **Figure 3—figure supplement 9A,D**. First five stripes are typically recognizable with $D_\theta = 0$, **Figure 3—figure supplement 9A**. With the increase in $D_\theta$, the local phase order decays more quickly and stripes are lost earlier, **Figure 3—figure supplement 9B–D**. Because ALD with a saturated dose of DAPT is around five in current experiments, we set $D_\theta = 0.0013$ throughout this study, **Figure 3—figure supplement 9B**. In resynchronization simulations with $\kappa(t) = \kappa_0 = 0.07$, both FRS and PLD do not depend on $D_\theta$ within its examined rage, **Figure 3—figure supplement 9E**. Hence, even if a different value of $D_\theta$ is used in resynchronization simulations, only a slight modification of the parameter values would be enough to better fit experimental FRS and PLD. Qualitative aspects of FRS and PLD in simulations do not depend on the value of $D_\theta$.

## Calculation of local phase order

To measure the level of synchrony at local domains along the anterior-posterior axis of the PSM, we introduce a local phase order parameter (**Shiogai and Kuramoto, 2003**). It is based on the Kuramoto phase order parameter (**Kuramoto, 1984**) with some modifications in computing average over cells to capture the presence of spatial phase waves in the PSM. For the left PSM $\Omega_l$, we first compute phase order for cells within a thin slice domain $\Omega_m(x) = [x + (m-1)\Delta x, \ x + m\Delta x] \times [0, \ 2r] \times [0, \ 2r]$ for $m = 1, 2, \ldots, M$ by:

$$Z_m(t,x)e^{\mathrm{i}\Psi_m(t,x)} = \frac{1}{n_m}\sum_{\mathbf{x}_j \in \Omega_m(x)} e^{\mathrm{i}\theta_j(t)}, \tag{16}$$

where $n_m$ is the number of cells within the domain $\Omega_m(x)$, **Figure 3—figure supplement 1A**. We set $\Delta x$ equal to the cell diameter $\Delta x = d_c$ so that **Equation (16)** measures the phase order for cells in same position in the anterior-posterior axis. $Z_m$ indicates the level of local synchrony of this slice domain. If the phases of cells in the domain are synchronized, $Z_m$ is close to one. In contrast, if they are not synchronized, $Z_m$ is close to zero. $\Psi_m$ indicates the mean phase of the cells in the slice domain.

This definition of $Z_m$ measures local order with a high spatial resolution along $x$-axis. However, only a few cells are present in each slice, introducing finite size fluctuations. Thus, we define the local phase order parameter at position $x$ by taking the average of $Z_m$ over a number $M$ of these domains, to smooth out small finite size fluctuations:

$$Z(t,x) = \frac{1}{M}\sum_{m=1}^{M} Z_m(t,x). \tag{17}$$

The segment length in the anterior-posterior direction in our parameter sets is $\sim 5\Delta x$. Therefore, we set $M = 5$ in the calculation of the local phase order **Equation (17)**. We calculate the local phase order parameter in a similar manner for the right PSM $\Omega_r$. In this case, the domain $\Omega_m(x)$ in **Equation (16)** is $\Omega_m(x) = [x + (m-1)\Delta x, \ x + m\Delta x] \times [2R, \ 2(R+r)] \times [0, \ 2r]$.

Note that computing local order in thin slices first as in **Equation (16)**, and then averaging as in **Equation (17)**, we can capture high local order values even in the presence of a phase gradient.

Computing a local order parameter directly in a thicker slab domain of width $5\Delta x$ would result in low values of the order parameter in the presence of a phase gradient as in the anterior PSM.

## Definition of a normal segment boundary in simulations

In computational simulations, we use the phase order parameter *Equation (17)* to define normal and defective segment boundaries. We first define a segment boundary by considering simulations for wild-type embryos untreated with DAPT. Such simulations are started from a synchronized initial condition. Then, we describe how to detect normal segment boundaries in resynchronization simulations started from random initial conditions.

In a untreated embryo where cells in the PSM are locally synchronized, kinematic phase waves can be observed across the tissue. In such embryos, the position of a new segment boundary is specified when a wave of gene expression arrives in the anterior end of the PSM. Namely, a position of a segment boundary is determined when the phase of cells near the anterior end of the PSM attains a certain value $\vartheta$, *Figure 3—figure supplement 1B,C*.

Based on this, we monitor the mean phase at the anterior end of the PSM $x_a$ to detect when a segment boundary position is set in simulations, *Figure 3—figure supplement 1A–C*. We compute the mean phase $\Psi_1(t, x_a)$ at position $x_a$ at time $t$ for $\Omega_1(x_a) = [x_a, \ x_a + \Delta x] \times [0, \ 2r] \times [0, \ 2r]$ for the left PSM and $\Omega_1(x_a) = [x_a, \ x_a + \Delta x] \times [2R, \ 2(R+r)] \times [0, \ 2r]$ for the right PSM by using *Equation (16)* ($m = 1$). As noted in *Equation (16)*, we set $\Delta x$ as the cell diameter $\Delta x = d_c$.

We then detect a time $\tau_i$ ($i = 1, 2, \ldots$) that satisfies:

$$\Psi_1(\tau_i, x_a) = \vartheta, \tag{18}$$

where $\vartheta$ is a constant that we set $\vartheta = 3\pi/2$ without loss of generality in this study, *Figure 3—figure supplement 1C*. For simulations for control embryos where DAPT is not added and, therefore, the level of synchrony is high, $\tau_i$ should be the time when the position of the segment boundary $i$ is determined, *Figure 3—figure supplement 1B,C*.

## Identification and numbering of defective segment boundaries

For resynchronization simulations starting from random initial conditions, we modify the above procedure as follows. After detecting the time $\tau_i$ when the mean phase of anterior cells becomes $\vartheta$, we check the local phase order parameter $Z(t, x_a)$ defined in *Equation (17)* to determine whether these cells can form a normal boundary, *Figure 3—figure supplement 1D*. We define that the anterior cells can form a normal boundary at time $\tau_i$ if:

$$Z(t, x_a) \geq Z_c \ \text{ for } \ \tau_i - T_a/2 \leq t \leq \tau_i - T_a/2 + \eta, \tag{19}$$

where $T_a$ is the period of oscillation at the anterior end of the PSM $x_a$ as described in the section "*PSM shortening*". Since $Z(\tau_i - T_a/2, x_a)$ is the average local phase order across nearly one segment length at $\tau_i - T_a/2$, it evaluates the integrity of the segment boundary and its neighboring inter-boundary regions. To suppress a false detection of a normal segment boundary caused by the fluctuation of $Z(t, x_a)$, we monitor $Z(t, x_a)$ in a short time interval with a window size $\eta$ in *Equation (19)*. We set $\eta = 4$ min in *Equation (19)*. By visual inspection of stripe patterns in simulations, we set $Z_c = 0.85$ for the recovery simulations throughout this paper, see *Figure 3F* and *Figure 3—figure supplement 1D,E*. Note that this threshold value is simply for detecting a normal segment boundary in simulations. It may be different from the critical value of the order parameter for normal segment boundary formation in actual embryonic tissues.

If *Equation (19)* is satisfied for $\tau_i$, we then specify the segment boundary number. Note that the subindex $i$ of $\tau_i$ does not specify the segment number in resynchronization simulations due to the fluctuation of the average phase $\Psi_1$ for earlier time when cells are not synchronized yet, *Figure 3—figure supplement 1D*. If the previous anterior cell population that satisfied $\Psi_1(\tau_{i-1}, x_a) = \vartheta$ at time $\tau_{i-1}$ ($\tau_{i-1} < \tau_i$) was also satisfied *Equation (19)* and numbered as segment boundary $j$, the current one is numbered as $j + 1$. If not, we infer the segment boundary number based on $\tau_i$ and anterior period $T_a$. We assign the current segment boundary with the expected segment number:

$$\begin{cases} [\![\tau_i/T_a]\!], & \text{if } [\![\tau_i/T_a]\!] - \tau_i/T_a < \Delta, \\ [\![\tau_i/T_a]\!] - 1, & \text{otherwise,} \end{cases} \tag{20}$$

where $\llbracket \chi \rrbracket$ represents rounding of real number $\chi$ to the closest integer value. Note that we assign the number $\llbracket \tau_i/T_a \rrbracket$ when the phases of anterior cells become $\vartheta$ slightly earlier, $\tau_i > T_a \llbracket \tau_i/T_a \rrbracket - \Delta T_a$ considering the fluctuation after resynchronization, *Figure 3—figure supplement 1F*. In this study, we set $\Delta = 0.3$ in *Equation (20)*. After detecting the normal segment boundaries and identifying their numbers in this way, we assign all the remaining segment boundaries to be defective.

## Definition of FRS and PLD in simulations

FRS is defined as $j_f - 1$ where $j_f$ is the smallest segment boundary number that was determined to be normal. The subtraction of 1 from $j_f$ is to match the definition of FRS for experimental data, see the section '*Whole-mount* in situ *hybridization and segmental defect scoring*'. PLD is defined as follows. We find the minimum segment boundary number $j_p$ above which all the segment boundaries, including $j_p$, are normal. Then, PLD is $j_p - 1$. In the example simulation shown in *Figure 3—figure supplement 1D,E*, FRS is 9 and PLD is 13.

## Calculation of phase vorticity

Vortices are regions in space where the phase values circulate from 0 to $2\pi$ around some point. Vorticity could be detected taking a closed path around cells and computing the accumulating change in the phase of neighboring cells around it. However, it is challenging to detect vorticity in a tissue where phase is not continuous in space, but only defined at points where cells are. Besides, there are phase fluctuations that can introduce local variations of phase change. Therefore, here we discretize the closed path in angular steps and average the phase over the resulting domains, *Figure 3—figure supplement 2*. The phase within these domains may grow linearly from 0 to $2\pi$ when one turns around a position close to the center of a vortex. Below, we describe the definition of vorticity that is shown in *Figure 3C*, *Figure 3—figure supplements 5* and *6* and *Figure 4—figure supplements 1*, *2* and *6*.

Vortex axis can have different spatial orientations. To detect vortices with different rotating axis, we set several planes in the space and compute phase vorticity at each plane. Then, we project phase vorticity on *x*-axis to obtain its trajectory along the anterior-posterior axis of the PSM. We first choose either the left or right PSM for the calculation of vorticity. Subsequently, we consider the four slices in the PSM, *Figure 3—figure supplement 2A,B*. These slices are two *z*-slices located at $z = 0$ μm and $z = 2r - 20$ μm (slices 1 and 2 and *Figure 3—figure supplement 2A*), and two *y*-slices located at $y = y_0$ μm and $y = y_0 + 2r - 20$ μm, (slices 3 and 4, *Figure 3—figure supplement 2B*), where $r$ is the radius of the PSM. For the left PSM, $y_0 = 0$ μm, while for the right PSM $y_0 = 2R$ μm where $R$ is the tailbud torus radius. The thickness of these four slices is 20 μm (~2 cell diameters, compare to the 50 μm of PSM diameter). We then project the phase values of cells in each slice to 2D planes, $\Pi^{(\alpha)}$ ($\alpha = 1, 2, 3, 4$). The two *x-y* planes $\Pi^{(1)}$ and $\Pi^{(2)}$ obtained by the projection of the two *z*-slices (slices 1 and 2) are used to detect vortices with a rotating axis parallel to *z*-axis (*Figure 3—figure supplement 2A,C*). For instance, the *x-y* plain $\Pi^{(1)}$ for the right PSM contains cells within the *z*-slice $[0, L_x] \times [2R, 2R + 2r] \times [0, 20]$. The two *x-z* planes $\Pi^{(3)}$ and $\Pi^{(4)}$ obtained by projection of the *y*-slices (slices 3 and 4) are used to detect vortices with a rotating axis parallel to *y*-axis, *Figure 3—figure supplement 2B*. We hardly observed phase vortices with the rotating axis parallel to *x*-axis in simulations. Therefore, we do not consider *y-z* planes in the calculation of vorticity.

Next, we set grids $(x_s, y_u)$ in the planes $\Pi^{(1)}$ and $\Pi^{(2)}$ where $x_s = x_a + s\Delta x$ ($s = 0, 1, 2, ...$) and $y_u = y_0 + u\Delta y$ ($u = 0, 1, 2, ...$) with the grid size $\Delta x$ and $\Delta y$. Similarly, we set grids $(x_s, z_u)$ to the planes $\Pi^{(3)}$ and $\Pi^{(4)}$ where $x_s = x_a + s\Delta x$ and $z_u = u\Delta z$. We chose $\Delta x = 5$ $\mu$m and $\Delta y = \Delta z = 2$ $\mu$m. For each grid point in the plane $\Pi^{(\alpha)}$, we compute vorticity $\psi$ as follows. Below, we explain the case of $\Pi^{(2)}$ with the grid $(x_s, y_u)$, *Figure 3—figure supplement 2A,C*. Same calculations were performed in the other planes as well.

We set a circular ring for the grid point $(x_s, y_u)$ in the plane:

$$\delta l \leq \sqrt{(x - x_s)^2 + (y - y_u)^2} < l + \delta l \tag{21}$$

as shown in *Figure 3—figure supplement 2D, E*. We set $\delta l = 5.5$ $\mu$m and $l = 14$ $\mu$m. The circular ring is subdivided into six domains $V_i$ ($i = 0, 1, 2, ..., 5$) by angles $\pi/3$ measured counterclockwise from the

x-axis. We then compute average phase $\bar{\theta}_i$ over cells within each subdomain $V_i$, $Z_i e^{i\bar{\theta}_i} = \sum_{k \in V_i} e^{i\theta_k}/n_i$ where $n_i$ is the number of cells in $V_i$. If there is no cell within one of the subdomains $V_i$ ($n_i = 0$), we do not compute the vorticity for the grid point $(x_s, y_u)$ and set $\psi = 0$.

To detect a vortex with clock-wise rotation, we permutate $\bar{\theta}_i$ based on their values (*Figure 3—figure supplement 2D, E*): $\hat{\theta}_0 = \bar{\theta}_k$ where $k = \arg\min\{\bar{\theta}_i: i = 0,...,5\}$, $\hat{\theta}_1 = \bar{\theta}_{k+1}$, $\hat{\theta}_2 = \bar{\theta}_{k+2}$,..., and $\hat{\theta}_5 = \bar{\theta}_{k+5}$, (mod 6). For a vortex with counter clock-wise rotation, we permutate $\bar{\theta}_i$ as: $\hat{\theta}_0 = \bar{\theta}_k$ where $k = \arg\min\{\bar{\theta}_i: i = 0,..., 5\}$, $\hat{\theta}_1 = \bar{\theta}_{k-1}$, $\hat{\theta}_2 = \bar{\theta}_{k-2}$,..., and $\hat{\theta}_5 = \bar{\theta}_{k-5}$, where a negative value of $k - j$ ($j = 1, 2, ..., 5$) should be replaced as $-1 \to 5$, $-2 \to 4$, ..., and $-5 \to 1$.

We assume that when a phase vortex is present near the grid point $(x_s, y_u)$, $\hat{\theta}_i$ for the grid point increases linearly with $i$, *Figure 3—figure supplement 2D, F*. To detect this linear increase of $\hat{\theta}_i$, we compute the correlation coefficient $\alpha$ defined as:

$$\alpha = \frac{1}{6\,\sigma_i \sigma_{\hat{\theta}}} \sum_{i=0}^{5} \left(i - \frac{5}{2}\right)\left(\hat{\theta}_i - \overline{\hat{\theta}}\right), \tag{22}$$

where $5/2 = \sum_{i=0}^{5} i/6$, $\overline{\hat{\theta}} = \sum_{i=0}^{5} \hat{\theta}_i/6$, $\sigma_i = \sqrt{\sum_{i=0}^{5}(i - 5/2)^2/6} = \sqrt{35/12}$, and $\sigma_{\hat{\theta}} = \sqrt{\sum_{i=0}^{5}\left(\hat{\theta}_i - \overline{\hat{\theta}}\right)^2/6}$. A value of the correlation coefficient $\alpha$ close to one means that the phase increases linearly along a perimeter of a circle, indicating the existence of a phase vortex, *Figure 3—figure supplement 2D,F*. If the correlation coefficient is larger than a threshold $\alpha \geq \alpha_0$, we consider that the phase value consistently increases along the circumference of the ring and rotates along the z-axis. In this case we define vorticity for the grid point as $\psi(x_s, y_u) = \left(\hat{\theta}_5 - \hat{\theta}_0\right)/2\pi$, *Figure 3—figure supplement 2F*. If $\alpha < \alpha_0$, we set $\psi(x_s, y_u) = 0$ to exclude false positive detection of a vortex by fluctuation of phase values. We used $\alpha_0 = 0.75$ throughout the article. After calculating vorticity for each grid point, we project $\psi(x_s, y_u)$ to x-axis. We use maximum projection, $\psi^{(2)}(x_s) = \max_{y_u} \psi(x_s, y_u)$, *Figure 3—figure supplement 2H*.

By performing same calculations for the remaining three planes, we obtain $\{\psi^{(1)}(x_s), \psi^{(2)}(x_s), \psi^{(3)}(x_s), \psi^{(4)}(x_s)\}$ for position $x_s$, top panel of *Figure 3—figure supplement 2I*. Finally, we take their maximum value and adopt it as the vorticity at the position $x_s$: $\psi^{(\max)}(x_s) = \max_{\alpha} \psi^{(\alpha)}(x_s)$, bottom panel of *Figure 3—figure supplement 2I*. For visualization, we make a density plot as a kymograph by using the data $\left(x_s, t, \psi^{(\max)}\right)$.

## Quantification of single and double defects

In both experiments and simulations, we sometimes observe that a defective segment boundary appears either only left or right side of an embryo, which we refer to as a single defect. We also observe another instance where left and right boundaries are both defective, which we call double defect. To examine whether the theory can account for the emergence of single defects in embryonic experiments, we compute the fraction $F_s$ of single defects defined below and compare it between simulations and experiment.

### Experimental data

We first counted the total number $N_t$ of defective segment boundary loci for an embryo. We then counted the number of single defects $N_s$ and computed the fraction $F_s = N_s/N_t$. When we compared the experimental data with simulation data, *Figure 4—figure supplement 8D*, we measured $N_t$ and $N_s$ after segment 9 that marks the onset of resynchronization, *Figure 1—figure supplement 1B*.

### Simulation data

We defined normal and defective segment boundaries based on the local phase order at the anterior end of the PSM $Z(t, x_a)$ as described in the previous section '*Definition of a normal segment boundary in simulations*'. For single realizations of simulation, we counted the total number of defective segment boundary loci $N_t$ and the number of single defects $N_s$ appeared posterior to the segment 9 as in experimental data. Then, we computed the fraction, $F_s = N_s/N_t$, *Figure 4—figure supplement 8D*.

## Implementation for numerical simulations

We solved *Equations (1) and (10)* with the Euler-Maruyama method with the time step for integration $\delta t = 0.01$ min. Custom simulation codes were written in C language (*Source code 1*). Videos of numerical simulations, calculations of local phase order and vorticity, and analysis of left-right segment boundary defects were done with custom Mathematica (Wolfram) codes (*Source code 1*).

## Acknowledgements

We thank the fish facility staff of the MPI-CBG, Dresden and Arianne Bercowsky, Gabriela Petrungaro, Sundar Naganathan, and Olivier Venzin for comments on the manuscript. Research was sponsored by JSPS KAKENHI grant number 17H05762, 19H04955, 19H04772 to KU; Ministry of Science and Technology, Taiwan (MOST 108–2311-B-019–001-MY3) to B-KL; SNSF Project funding division III (31003A_176037) and Wellcome Trust Senior Research Fellowship in Basic Biomedical Science (WT098025MA) to AO; European Research Council Starting Independent Research Grant (ERC-2007-StG: 207634) and the Francis Crick Institute to B-KL and AO; ANPCyT PICT 2012 1954, PICT 2013 1301, PICT 2017 3753 to LGM, FOCEM-Mercosur (COF 03/11) to IBioBA; and JSPS Short Term Grant S17064 to KU and LGM.

## Additional information

### Funding

| Funder | Grant reference number | Author |
|---|---|---|
| Japan Society for the Promotion of Science | KAKENHI grant number 17H05762 | Koichiro Uriu |
| Japan Society for the Promotion of Science | KAKENHI grant number 19H04955 | Koichiro Uriu |
| Japan Society for the Promotion of Science | KAKENHI grant number 19H04772 | Koichiro Uriu |
| Ministry of Science and Technology, Taiwan | MOST 108-2311-B-019-001-MY3 | Bo-Kai Liao |
| SNSF | 31003A_176037 | Andrew C Oates |
| Wellcome Trust | WT098025MA | Andrew C Oates |
| European Research Council | ERC-2007-StG: 207634 | Bo-Kai Liao<br>Andrew C Oates |
| Francis Crick Institute | | Bo-Kai Liao<br>Andrew C Oates |
| Agencia Nacional de Promoción Científica y Tecnológica | PICT 2012 1954 | Luis G Morelli |
| Agencia Nacional de Promoción Científica y Tecnológica | PICT 2013 1301 | Luis G Morelli |
| Agencia Nacional de Promoción Científica y Tecnológica | PICT 2017 3753 | Luis G Morelli |
| FOCEM-Mercosur | COF 03/11 | Luis G Morelli |
| Japan Society for the Promotion of Science | Short Term Grant S17064 | Koichiro Uriu<br>Luis G Morelli |

The funders had no role in study design, data collection and interpretation, or the decision to submit the work for publication.

### Author contributions

Koichiro Uriu, Conceptualization, Software, Formal analysis, Funding acquisition, Investigation, Visualization, Methodology, Writing - original draft, Writing - review and editing; Bo-Kai Liao, Conceptualization, Resources, Funding acquisition, Validation, Investigation, Visualization, Writing - original

draft, Writing - review and editing; Andrew C Oates, Conceptualization, Supervision, Funding acquisition, Visualization, Writing - original draft, Writing - review and editing; Luis G Morelli, Conceptualization, Formal analysis, Funding acquisition, Visualization, Methodology, Writing - original draft, Writing - review and editing

### Author ORCIDs
Koichiro Uriu https://orcid.org/0000-0003-1802-2470
Bo-Kai Liao https://orcid.org/0000-0003-4742-4487
Andrew C Oates https://orcid.org/0000-0002-3015-3978
Luis G Morelli https://orcid.org/0000-0001-5614-073X

### Ethics
Animal experimentation: Zebrafish experimentation was carried out in strict accordance with the ethics and regulations of the Saxonian Ministry of the Environment and Agriculture in Germany under licence Az. 74-9165.40-9-2001, and the Home Office in the United Kingdom under project licence PPL No. 70/7675.

### Decision letter and Author response
Decision letter https://doi.org/10.7554/eLife.61358.sa1
Author response https://doi.org/10.7554/eLife.61358.sa2

## Additional files
### Supplementary files
- Source code 1. Custom C and Mathematica codes for simulation, visualization, and data analysis.
- Supplementary file 1. Parameter values used in *Figures 2* and *3*.
- Supplementary file 2. Parameter values used in *Figure 4* and figure supplements.
- Transparent reporting form

### Data availability
All data generated or analyzed during this study are included in the manuscript and supporting files.

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

## Appendix 1

### Segment statistics from the spatial distribution of defective segments

The distribution of defects along the embryonic axis can be parametrized in different ways. Here, we introduce complementary pictures, one relies on the fraction of left and right defects, and the other on fractions of single and double defects. Using a random defect hypothesis we show how to relate these pictures with probability theory. Taking into account the spatial distribution of defects we compute ALD, PLD and FRS from these statistics, and we predict and test the values of the fraction of single defects per embryo.

### Segment state variables

We introduce a state variable that accounts for the presence or absence of a defective segment $S_{ik}(x)$, where $i$ labels the embryo, $k = \{l, r\}$ labels the side of the embryo and $x$ is the segment locus along the axis. We consider $N$ embryos with $M + 1$ segment boundaries, so $i = 1, 2, ..., N$ and $x = 0, 1, ..., M$. The state variable $S_{ik}(x)$ takes the values zero and one depending on whether the segment is normal or defective.

### Segment defect distribution on both sides of the embryo

When a segment locus is defective on the left (right) side of the embryo we call it a left (right) defect independently of the state of the other side. We define left and right defect distributions taking the population average of segment state variables,

$$p_l(x) = \langle S_{il}(x) \rangle = \frac{1}{N} \sum_{i=1}^{N} S_{il}(x), \tag{23}$$

and

$$p_r(x) = \langle S_{ir}(x) \rangle = \frac{1}{N} \sum_{i=1}^{N} S_{ir}(x). \tag{24}$$

These spatial distributions of defective segments on the left and right sides of embryos are very similar, for different DAPT washout timing, *Figure 4—figure supplement 7A*. This agreement between left and right distributions supports the assumption $p_l(x) = p_r(x) = p(x)$. In the following sections, we use probability theory to compute ALD, PLD and FRS from this spatial distribution of defective segments.

### Probabilistic calculation of ALD

Let $q_a(x_a)$ be the probability to find an ALD at position $x_a$. $q_a(x_a)$ can be expressed in terms of $p(x)$ as:

$$q_a(x_a) = \begin{cases} p(x_a) & x_a = 0 \\ \prod_{\xi=0}^{x_a-1} (1 - p(\xi)) \times p(x_a) & x_a > 0. \end{cases} \tag{25}$$

The first factor $\prod_{\xi=0}^{x_a-1}(1 - p(\xi))$ in the second line represents the probability of normal segment boundaries from position $x = 0$ to $x = x_a - 1$. The second factor $p(x_a)$ is the probability of a defective segment boundary at position $x = x_a$. The resulting ALD distribution $q_a(x_a)$ presents a clear peak at the onset of the defective region, *Figure 4—figure supplement 7B*. The ALD is then calculated as the mean value for this probability distribution,

$$\text{ALD} = \sum_{x_a=0}^{M} x_a \cdot q_a(x_a). \tag{26}$$

## Probabilistic calculation of PLD

Let $q_p(x_p)$ be the probability of PLD at position $x_p$. It can be written as:

$$q_p(x_p) = \begin{cases} p(x_p) \times \prod_{\xi=x_p+1}^{M} (1-p(\xi)) & x_p < M \\ p(x_p), & x_p = M. \end{cases} \tag{27}$$

The first factor $p(x_p)$ in the first line is the probability of a defective boundary at $x_p$. The second factor $\prod_{\xi=x_p+1}^{M}(1-p(\xi))$ represents the probability that all the remaining segment boundaries posterior to $x_p$ are normal. The resulting PLD distribution $q_p(x_p)$ peaks at the end of the defective region, *Figure 4—figure supplement 7B*. The PLD can be written as the mean value for this distribution,

$$\text{PLD} = \sum_{x_p=0}^{M} x_p \cdot q_p(x_p). \tag{28}$$

## Probabilistic calculation of FRS

Occurrence of a recovered segment is conditioned to the previous occurrence of defective segments. In this study, we measure the FRS after the desynchronization phase. The desynchronization phase is determined based on the distribution of defective segments, *Figure 1—figure supplement 1B*. Suppose that the desynchronization phase ends by the formation of segment $s_d$. We define FRS as the first normal segment after $s_d - 1$. This is the definition we use to measure the embryonic FRS. With this definition of FRS, the probability $q_f(x_f)$ of the first normal segment boundary at locus $x_f$ is:

$$q_f(x_f) = \begin{cases} 1-p(x_f), & x_f = s_d \\ \prod_{\xi=s_d}^{x_f-1} p(\xi) \times (1-p(x_f)), & x_f > s_d. \end{cases} \tag{29}$$

The first factor of the second line represents the probability for all the segment boundaries between $s_d$ and $x_f - 1$ to be defective. The second factor is the probability for a normal segment boundary to form at $x_f$.

To compute $q_f(x_f)$, we set $s_d = 9$ as in the main text, *Figure 1—figure supplement 1B*. The resulting distribution $q_f(x_f)$ has a peak that precedes that of PLD and partly overlaps with it, *Figure 4—figure supplement 7B*. From these results, the FRS then can be expressed as

$$\text{FRS} = \sum_{x_f=s_d}^{M} x_f \cdot q_f(x_f) - 1. \tag{30}$$

In *Equation (30)*, we subtract one because FRS is defined with the anterior boundary of the first normal segment after $s_d - 1$, see definition of FRS for experimental data in Materials and methods. $q_a$, $q_p$ and $q_f$ calculated with *Equations (25), (27) and (29)* agree well with the direct measurement of these distributions, *Figure 4—figure supplement 7B*. Furthermore, expressions obtained for ALD, PLD and FRS from the spatial distribution of defects $p(x)$, *Equations (26), (28) and (30)*, are in very good agreement with direct measurements of these quantities, *Figure 4—figure supplement 7C*.

## The spatial distribution of single and double defects

In a complementary framework, we introduce double defects, single defects and normal segments. Double defects occur when both sides of the embryo at a given locus $x$ are defective. Single defects occur when at a given locus there is a defect either left or right, but not on the other side. Normal segments have no defects on either side. Taking population averages, we can define the double defect distribution

$$p_d(x) = \langle S_{il}(x)\, S_{ir}(x) \rangle = \frac{1}{N}\sum_{i=1}^{N} S_{il}(x)\, S_{ir}(x), \tag{31}$$

the single defects distribution

$$p_s(x) = \frac{1}{N}\sum_{i=1}^{N} \left( S_{il}(x)\,(1 - S_{ir}(x)) + (1 - S_{il}(x))\, S_{ir}(x) \right), \tag{32}$$

and the normal segments distribution

$$p_n(x) = \frac{1}{N}\sum_{i=1}^{N} (1 - S_{il}(x))(1 - S_{ir}(x)). \tag{33}$$

These distributions can be related to the left/right framework introduced above. Key to this is that the two sides of the anterior PSM are physically unconnected, separated by another tissue –the notochord. It has been shown that interfering with Retinoic Acid, which controls somitogenesis bilateral symmetry by shielding asymmetric cues, results in asymmetric left/right segmentation and clock waves (*Vermot et al., 2005*). This indicates that segmentation clock oscillations are independent in the left and right sides of the PSM. In the physical model, we tacitly assume this independence since there is no coupling between oscillators on one side and the other. A consequence of this left/right independence should be a vanishing covariance of the segment defect variables at opposite sides of the embryo (*Gardiner, 2009*),

$$\langle S_{il}(x)\, S_{ir}(x) \rangle - \langle S_{il}(x) \rangle \langle S_{ir}(x) \rangle = 0. \tag{34}$$

The two terms in this covariance can be written using the segment defect distributions of the two frameworks,

$$p_d(x) = p_l(x) p_r(x). \tag{35}$$

Similarly, we obtain for single defects

$$p_s(x) = p_l(x)\,(1 - p_r(x)) + (1 - p_l(x))\, p_r(x), \tag{36}$$

and for normal segments

$$p_n(x) = (1 - p_l(x))(1 - p_r(x)). \tag{37}$$

We can further simplify these expressions with the assumption $p_l(x) = p_r(x) = p(x)$. The axial distribution of double defects $p_d(x)$ gradually grows to a plateau at the onset of the defective region and then decays at its end, while the distribution of single defects $p_s(x)$ peaks both at the onset of the defective region and at its end, *Figure 4—figure supplement 8A*. The good agreement observed between direct measurement of $p_d(x)$ and $p_s(x)$ and results obtained from $p(x)$ together with probabilistic arguments, provides a test for the vanishing of the covariance and left/right independence, *Figure 4—figure supplement 8A*.

## Fraction of single defects from the axial distribution of defects

As a further test of the reach of the defect distribution $p(x)$, we use it to compute the fraction of single defects in a population of embryos. We first count the number of double and single defects in a given embryo $i$

$$N_{id} = \sum_{x=0}^{M} S_{il}(x) S_{ir}(x), \tag{38}$$

and

$$N_{is} = \sum_{x=0}^{M} (S_{il}(x)(1-S_{ir}(x)) + (1-S_{il}(x))S_{ir}(x)). \tag{39}$$

The total number of defects in embryo $i$ is $N_{it} = N_{is} + N_{id}$. Then, we take the population average of these quantities. The average number of double defects in the population is

$$\langle N_{id} \rangle = \frac{1}{N}\sum_{i=1}^{N} N_{id} = \frac{1}{N}\sum_{i=1}^{N}\sum_{x=0}^{M} S_{il}(x)\,S_{ir}(x) = \sum_{x=0}^{M}\frac{1}{N}\sum_{i=1}^{N} S_{il}(x)\,S_{ir}(x), \tag{40a}$$

so

$$\langle N_{id} \rangle = \sum_{x=0}^{M}\langle S_{il}(x)\,S_{ir}(x)\rangle = \sum_{x=0}^{M} p_d(x) = \sum_{x=0}^{M} p_l(x)p_r(x), \tag{40b}$$

using left/right independence. Similarly, the average number of single defects $\langle N_{is}\rangle$ is

$$\langle N_{is}\rangle = \sum_{x=0}^{M} p_s(x) = \sum_{x=0}^{M}(p_l(x)(1-p_r(x)) + (1-p_l(x))\,p_r(x)). \tag{41}$$

We define the fraction of single to total defects in an embryo

$$F_{is} = N_{is}/N_{it}, \tag{42}$$

and its population average

$$F_s = \langle F_{is}\rangle = \langle N_{is}/N_{it}\rangle. \tag{43}$$

If the coefficient of variation of the total number of defects in the population is small, we can approximate

$$F_s = \langle N_{is}/N_{it}\rangle \approx \langle N_{is}\rangle/\langle N_{it}\rangle, \tag{44a}$$

so

$$F_s \approx \frac{\langle N_{is}\rangle}{\langle N_{it}\rangle} = \frac{\sum_{x=0}^{M}(p_l(x)(1-p_r(x)) + (1-p_l(x))\,p_r(x))}{\sum_{x=0}^{M}(1-(1-p_l(x))(1-p_r(x)))}. \tag{44b}$$

Using the left/right symmetry of defect distributions $p_l(x) = p_r(x) = p(x)$, we can further simplify this expression

$$F_s \approx \frac{\langle N_{is}\rangle}{\langle N_{it}\rangle} = \frac{2\sum_{x=0}^{M} p(x)(1-p(x))}{\sum_{x=0}^{M}\left(1-(1-p(x))^2\right)}. \tag{45}$$

The good agreement observed between this probabilistic calculation and direct measurement of $F_s$ counting single defects in individual embryos, *Figure 4—figure supplement 8C*, suggests that the recovery of segments after DAPT washout occurs independently between left and right sides of the PSM.

Finally, the physical model of the PSM reproduces both the single and double defect distributions, *Figure 4—figure supplement 8B*, and the dependence of $F_s$ on DAPT washout timing, *Figure 4—figure supplement 8D*.

## Conclusion

Taken together, these results suggest that knowledge of $p(x)$ is enough to compute some of the key observables that we use to quantify segment recovery, such as ALD, PLD, FRS and $F_s$. Together with the result that the physical model of the PSM reproduces $p(x)$ at different washout timings, *Figure 4F*, this is evidence for the breadth of the physical model results and predictions.

