## [Decision Letter]

**Acceptance summary:**

This paper represents a step forward in combining quantitative experiments and modeling in order to explain the mechanism behind complex dynamic phenomena in a terms of the underlying molecular and genetic interactions.

**Decision letter after peer review:**

Thank you for submitting your article "From local resynchronization to global pattern recovery in the zebrafish segmentation clock" for consideration by *eLife*. Your article has been reviewed by three peer reviewers, one of whom is a member of our Board of Reviewing Editors, and the evaluation has been overseen by Didier Stainier as the Senior Editor. The reviewers have opted to remain anonymous.

The reviewers have discussed the reviews with one another and the Reviewing Editor has drafted this decision to help you prepare a revised submission.

Summary:

All three reviewers agreed that this work makes a valuable contribution to the field and should be published with revisions.

The paper presents an elegant yet complex mathematical and computational model for this embryo segment formation that quantitatively reproduces phenotypes observed after pharmacological inhibition of Notch signaling in Zebrafish embryos. The zebrafish segmentation provides an important research paradigm for developmental processes where tissue patterning and growth are tightly coupled. Intriguingly, the model integrates many different parameters at different levels of organization, i.e. from local cell-cell coupling to tissue level parameters, such as shrinking rate of the presomitic mesoderm. The agreement between simulations and experiments is excellent. This multi-scale model allows the authors to make several non-intuitive predictions, which can also guide future experiments.

In particular, the authors find evidence for vortices formed by the synchronizing genetic oscillators under certain conditions, leading to a large-scale and semi-persistence perturbation of the natural traveling-wave front pattern. Understanding such collective patterns is important to properly interpret mutant phenotypes, furthermore, to understand the critical parameter choices required to stably produce the desired outcome under WT conditions. This work will likely help understanding error correction and robustness in many developmental / multicellular contexts.

The reviewers find the paper overall well written and the videos especially helpful to visualize the dynamics. Each of the reviewers has some suggestions that will help to improve this paper further.

The main points summarized are (see full points below):

• The Abstract and Discussions should be sharpened. The authors should re-evaluate how they summarize the main insights of this paper, also to make this message more obviously relevant to researchers in related fields. Some of the writing should be improved to make the paper more accessible especially to non-specialists in the field.

• Overall the paper should be highlighted as a primarily theoretical contribution (with some experimental support) – as otherwise readers will wonder why many suggestive experiments have not been carried out. At various points it is also not directly obvious whether the authors talk about experimental or theoretical results.

• Do the authors have any additional evidence (in their own lab / unpublished by other groups / from the literature) that such vortices actually occur?

• The reason and potential impact on results for some of the model assumptions (simplifications) have not been explicitly addressed, such as changes in segment size, time delays, or the role of left-right communication. And how does the model exactly compare to already published models?

• The authors should evaluate whether additional, more direct insights can be derived from the model (see suggestions). For example, can we more directly understand what accounts for changes in vortex size? Can we quantitatively understand such features in a more direct, intuitive way – beyond "it comes out of a complex simulation"?

Reviewer #1:

The zebrafish segmentation provides an important research paradigm for developmental processes where tissue patterning and growth are tightly coupled. This paper combines quantitative experimental and novel theoretical approaches, which promises quantitative and deeper insights into this system. In particular, the authors find evidence for vortices formed by the synchronizing genetic oscillators under certain conditions, leading to a large-scale and semi-persistence perturbation of the natural traveling-wave front pattern. Understanding such collective patterns is important to properly interpret mutant phenotypes, furthermore, to understand the critical parameter choices required to stably produce the desired outcome under WT conditions. The results are likely of high relevance for understanding error correction and robustness in many developmental / multicellular contexts.

I recommend sharpening some of the discussions, furthermore, to evaluate whether additional, more direct insights can be derived from the model. This provides the opportunity to be more helpful to a wider readership.

Reviewer #2:

In this manuscript a theoretical model for embryo segment formation is presented that quantitatively reproduces phenotypes observed after pharmacological inhibition of Notch signaling in Zebrafish embryos. Intriguingly, the model integrates many different parameters at different levels of organization, i.e. from local cell-cell coupling to tissue level parameters, such as shrinking rate of the presomitic mesoderm. This multi-scale model allows the authors to make several non-intuitive predictions, which can guide future experiments. I see here a great value of this theoretical work.

Points to discuss:

1) I think one point of criticism that likely will be raised is the fact that several (most?) of these predictions are not tested experimentally, also because there is no straightforward way to tune parameters such as tissue growth, cell motility with the specificity needed.

To address this criticism, I would recommend to label and present the work more explicitly as theoretical work and hence I would move the key parts of the theory to the main text, rather than (mis-?) placing these in the Materials and methods section… The theoretical model is a key result that should be presented as such. In addition, I would be explicit about the fact that several predictions are not tested within this paper but rather, this work serves to guide follow up studies, in different systems. I suggest, in order to make this very accessible, to prepare a list of predictions and detail whether or not these have been tested already, it will serve as guide for future experimentation which than refer back to this original theoretical work.

2) One central prediction is the occurrence of vortices (phase vortices) during the re-synchronization process. While, as pointed out, vortices occur in a wide range of oscillatory systems and hence per se not might be diagnostic, the demonstration would obviously be crucial to support the validity of the presented model. Given that the authors of this manuscript have access to refined imaging setup to quantify oscillations in zebrafish embryos, I would have expected a more decisive answer. To state this data should "soon" be available sounds a bit vague (what means soon?), have these vortices be observed after DAPT washout, has it been attempted?

3) Similarly, as indeed phase vortex can arise in many systems and as the authors predict that "these structures [vortices] will form also in mammalian PSM tissue culture systems" the obvious question is whether indeed there is experimental evidence for the presence of vortices during synchronization in mouse embryo re-synchronization experiments, or stem cells systems (mouse/human, gastruloids). In all these, oscillation dynamics have been quantified in real-time. Have the authors reached out to those groups to query the data? I think this would be in the spirit of this work, i.e. make theoretical progress first and then query the experimental data available in the field.

4) The rationale underlying some model assumptions is not evident:

a) for instance, in the case of shrinking PSM, the authors assume that segment size remains constant. To achieve this in their model, they reduce advection speed to compensate for boundary Xa movement. However, experimental evidence has recently been published using Zebrafish embryos providing evidence that while PSM size is reduced, segment size do not remain of constant size but rather, show a scaling behaviour (Simsek et al., 2018). Can the authors address this point in their model to relate to previous findings?

b) In their model, they do not include time-delays for intercellular communication, while stating that these "play important roles in setting the period of collective rhythms and synchronization". How does the exclusion of this key feature affect the conclusions?

Reviewer #3:

The manuscript by Uriu and colleagues addresses the process of resynchronization and segment recovery in somitogenesis in zebrafish embryos. On the one hand, it shows that recovery after early washout of DAPT drives the intermingling of defective and normal segments in zebrafish embryos. On the other hand, it presents an elegant yet complex mathematical and computational model to propose how this intermingling arises. The agreement between simulations and experiments is excellent. The manuscript is very well written, all data is very clear and the conclusions are very well supported. Videos are especially helpful to visualize the dynamics. I believe this is an excellent and elegant work that deserves publication in *eLife*.

A major comments to improve the manuscript is:

In the simulations, normal segments arise when the most posterior part of the tailbud spontaneously becomes synchronized. It is then when waves propagate from posterior to anterior, symmetrically on left and right sides, without any spiral. If I am correct, it is the geometry of the PSM and tailbud (two cylinders connected by a toroidal shape), together with the gradient of frequencies, what enables that "wave initiation" becomes ultimately localized within the wide region at the posterior end of the tailbud. During intermingling, wave initiation starts at other locations, more locally, within the PSM. I would suggest to discuss on the relevance of the overall geometry of the PSM+tailbud, that involves left-right symmetry, to reach re-synchronization. In the absence of left-right communication (e.g. in simulations where there is only one tubular shape with no toroidal mimicking the tailbud, for instance) I would expect that recovery of normal segmentation would take much longer, if it happens, and intermingling will be affected.

---

## [Author Response]

The main points summarized are (see full points below):• The Abstract and Discussions should be sharpened. The authors should reevaluate how they summarize the main insights of this paper, also to make this message more obviously relevant to researchers in related fields. Some of the writing should be improved to make the paper more accessible especially to non-specialists in the field.

We modified the Abstract following reviewer 1’s comments, and thoroughly revised the Discussion following multiple points from reviewer 1 and reviewer 2, in particular expanding on some ideas that were previously expressed in a more technical language, aiming to make them more generally accessible.

• Overall the paper should be highlighted as a primarily theoretical contribution (with some experimental support) – as otherwise readers will wonder why many suggestive experiments have not been carried out. At various points it is also not directly obvious whether the authors talk about experimental or theoretical results.

In the Abstract, we have indicated explicitly the approach and results coming from the theory. We re-wrote all the subsection titles to unambiguously indicate whether each part concerned experiment, theory, or had contributions from both.

Following reviewer 2’s suggestion, we have moved key equations defining the theory into the main text at the place where the model is introduced. In addition, there are new paragraphs on the limitations and assumptions of the model, and its relationship to previous approaches, which appear primarily in the Discussion. Together, we hope that these changes have helped to highlight the central role of theory in the current manuscript.

• Do the authors have any additional evidence (in their own lab / unpublished by other groups / from the literature) that such vortices actually occur?

We are currently attempting to image the dynamics of re-synchronization in transgenic zebrafish in our lab, but this is challenging – not least because of the Covid-19 confinements – and efforts are still ongoing. We have contacted several experimental labs to ask whether they see spirals or other signs of vortex formation, and their responses have been cautiously positive. The vortex-like wave dynamics already seen in Hubaud et al., 2017, Movie S3, are indeed reproducible in different labs, but the experiments to date were not designed to either produce or detect vortices. Thus, it turns out the existing data are not suitable for a straightforward or definitive analysis (individual cells are not trackable, for example). We expand and discuss these points extensively below.

• The reason and potential impact on results for some of the model assumptions (simplifications) have not been explicitly addressed, such as changes in segment size, time delays, or the role of left-right communication. And how does the model exactly compare to already published models?

The reason and potential impact on results of the model assumptions of changes in segment size, time delays and left-right communication have all been explicitly addressed now. Furthermore, we have described the lineage of the model so that the reader can see exactly from which models this one extends, as well as included comparisons to other types of models.

• The authors should evaluate whether additional, more direct insights can be derived from the model (see suggestions). For example, can we more directly understand what accounts for changes in vortex size? Can we quantitatively understand such features in a more direct, intuitive way – beyond "it comes out of a complex simulation"?

We identified two additional insights that could be derived from the model. At the suggestion of reviewer #1, we have characterized the size of defects in both embryo and simulation, and found good agreement – new panels G and H in Figure 3 and Figure 3—figure supplement 3. This allows us to be more explicit about the cause of the segmental defects, with an intuitive explanation being that the intermingled defects are caused by intermittently released vortices of relatively small and consistent size passing through the anterior boundary. Another insight of the model, which we had not previously explained, is the fundamental difference between re-synch (vortices are seeded by local phase differences and driven by coupling), and de-synch (starts from a near-uniform spatial phase in the absence of coupling). We hope that these new explanations give the reader some direct and intuitive insight into the basic process at work.

We note that two reviewers commented on changes in vortex size, perhaps motivated by defect size distributions, as above, but also looking at the kymographs of vorticity. However, we believe there may be some misunderstanding about vortex size since the vorticity index, by definition, measures the presence of a vortex core passing through a specified circular region, but does not say anything about the edge of the vortex. The apparent “width” of a vortex in a kymograph is also bounded by the diameter of the circular region, which in our case was set to the width of the tissue. So, this index is designed to identify the location of a vortex, but not to measure its spatial extent. We discuss this in detail below, and have modified the main text to clarify this point.

Reviewer #1:The zebrafish segmentation provides an important research paradigm for developmental processes where tissue patterning and growth are tightly coupled. This paper combines quantitative experimental and novel theoretical approaches, which promises quantitative and deeper insights into this system. In particular, the authors find evidence for vortices formed by the synchronizing genetic oscillators under certain conditions, leading to a large-scale and semi-persistence perturbation of the natural traveling-wave front pattern. Understanding such collective patterns is important to properly interpret mutant phenotypes, furthermore, to understand the critical parameter choices required to stably produce the desired outcome under WT conditions. The results are likely of high relevance for understanding error correction and robustness in many developmental / multicellular contexts.I recommend sharpening some of the Discussions, furthermore, to evaluate whether additional, more direct insights can be derived from the model. This provides the opportunity to be more helpful to a wider readership.

We have extensively revised and expanded the Discussion, to make what we consider the key points more focused transparent, in part following the critique and questions of the reviewers.

Reviewer #2:In this manuscript a theoretical model for embryo segment formation is presented that quantitatively reproduces phenotypes observed after pharmacological inhibition of Notch signaling in Zebrafish embryos. Intriguingly, the model integrates many different parameters at different levels of organization, i.e. from local cell-cell coupling to tissue level parameters, such as shrinking rate of the presomitic mesoderm. This multi-scale model allows the authors to make several non-intuitive predictions, which can guide future experiments. I see here a great value of this theoretical work.Points to discuss:1) I think one point of criticism that likely will be raised is the fact that several (most?) of these predictions are not tested experimentally, also because there is no straightforward way to tune parameters such as tissue growth, cell motility with the specificity needed.To address this criticism, I would recommend to label and present the work more explicitly as theoretical work and hence I would move the key parts of the theory to the main text, rather than (mis-?) placing these in the Materials and methods section. The theoretical model is a key result that should be presented as such. In addition, I would be explicit about the fact that several predictions are not tested within this paper but rather, this work serves to guide follow up studies, in different systems. I suggest, in order to make this very accessible, to prepare a list of predictions and detail whether or not these have been tested already, it will serve as guide for future experimentation which than refer back to this original theoretical work.

We have made changes to the Abstract to better reflect the theoretical aspects of the work, we have modified the section subtitles to reflect the contribution of theoretical or experimental approaches to each section, and we have substantially expanded the discussion of several key aspects of the theory in the Discussion section.

We have also now moved the two key equations describing cell mechanics and the phase oscillators to the main text, placing them in the results where the model is first introduced and briefly discussing each term in these equations.

Furthermore, we have rewritten the final two paragraphs of the Discussion to highlight the predictions of the theory that have already been tested in this paper (spatial defect distribution, left-right independence of defects, and the size of defects [as suggested by reviewer 1]), and state those that are still to be tested (existence of vortices, vortices cause intermingled defects). This is followed by several sentences on the practical requirements and potential challenges of directly observing the defects in vivo, see point 2 below.

2) One central prediction is the occurrence of vortices (phase vortices) during the re-synchronization process. While, as pointed out, vortices occur in a wide range of oscillatory systems and hence per se not might be diagnostic, the demonstration would obviously be crucial to support the validity of the presented model. Given that the authors of this manuscript have access to refined imaging setup to quantify oscillations in zebrafish embryos, I would have expected a more decisive answer. To state this data should "soon" be available sounds a bit vague (what means soon?), have these vortices be observed after DAPT washout, has it been attempted?

Soon is indeed a bit vague, and it’s not necessary to give this time qualification, so we removed the word. We are indeed currently imaging the dynamics of re-synchronization in transgenic zebrafish in our own lab, and this is challenging – not least because of the Covid-19 confinements this year – and still ongoing. Some of the challenges include: tracking the tail of the embryo as it elongates and displaces on the imaging stage, tracking the cells in the densely packed tissue, and reliably extracting the phase from gene expression signal with large amplitude fluctuations. We have made progress in all these areas, but the largest delay has come from needing to generate a new oscillating reporter line with reduced amplitude fluctuations, which we believe is now closer to the endogenous level. Nevertheless, we have seen rotating patterns in experiments that use this new segmentation clock transgene, arising from a different perturbation, so we anticipate that technically we should be able to identify and characterise vortices during re-synchronization so as to be able to definitively confirm or deny the predictions of our model. We have now included some more information in the Discussion about the practical requirements for seeing such vortices.

3) Similarly, as indeed phase vortex can arise in many systems and as the authors predict that "these structures [vortices] will form also in mammalian PSM tissue culture systems" the obvious question is whether indeed there is experimental evidence for the presence of vortices during synchronization in mouse embryo re-synchronization experiments, or stem cells systems (mouse/human, gastruloids). In all these, oscillation dynamics have been quantified in real-time. Have the authors reached out to those groups to query the data? I think this would be in the spirit of this work, i.e. make theoretical progress first and then query the experimental data available in the field.

As mentioned in the brief overview to the editor, the data from the literature that we point out in the discussion as potentially harboring such vortices, such as the PSM cultures, indeed show complex wave dynamics, consistent with a vortex (for example, see Hubaud et al., 2017, Movie S3). We have contacted several experimental labs to ask whether they see spirals or other signs of vortex formation, and their responses have been cautiously positive – the complex wave dynamics mentioned above are indeed reproducible in different labs. However, when looking at the current form of the data, it becomes clear that the experiments to date were not designed to either produce or detect vortices, and are not suitable for a straightforward analysis (individual cells can’t be tracked, for example). So, although this is a very promising avenue, it will be tackled through new projects, where some of these groups are now actively looking for the vortices. We hope our manuscript will in addition spur other groups with PSM cultures or similar in vitro systems to search for their existence and characteristics.

4) The rationale underlying some model assumptions is not evident:a) for instance, in the case of shrinking PSM, the authors assume that segment size remains constant. To achieve this in their model, they reduce advection speed to compensate for boundary Xa movement. However, experimental evidence has recently been published using Zebrafish embryos providing evidence that while PSM size is reduced, segment size do not remain of constant size but rather, show a scaling behaviour (Simsek et al., 2018). Can the authors address this point in their model to relate to previous findings?

The shorter lengths of segments forming in the tail have been recognized for many years. Likewise, the correlated shortening of PSM and segment length is not disputed, but its mechanism is not well understood. This segment length is a measurement that is made at the instant of somite formation, and lengths change thereafter as the embryo grows and metamorphoses into a larva, and in the long run, these early segment differences are essentially equalized in the adult body skeleton.

Our data comes from an assay chosen for its high sensitivity to boundary defect detection, and this is measured at an embryological stage a day after segmentation has finished. Furthermore, this assay has been widely used already, and thus connects to existing literature. Thus, we didn’t measure or compare the absolute lengths of any segments in our work, rather we counted the number of segments – in other words, we estimated the time duration of any defective or recovery behavior. Therefore, we did not model changing segment length in the current work.

The qualitative conclusions of our study are not changed by a changing segment length, as this would have no direct bearing on the existence of vortices and intermingled segment defects, but the quantitative agreement between data and model for the posterior / later time intervals could be influenced by the underlying processes that give rise to shorter segments in the tail, for example a slower wavefront velocity. Any potential role for changing segment length remains an interesting open question, and clearly this theoretical framework could be used to investigate this, in combination with timelapse analysis of defect generation during ongoing somitogenesis. However, this is a large, heavily experimental project, and will have to remain beyond the scope of the current work.

To place this assumption in context, we have now better motivated the use of the marker in the Results, and discussed the assumption and implications of constant segment length in paragraph four of the Discussion.

b) In their model, they do not include time-delays for intercellular communication, while stating that these "play important roles in setting the period of collective rhythms and synchronization". How does the exclusion of this key feature affect the conclusions?

In this paper delays were neglected to reduce computational complexity and time. In preliminary simulations of the segmentation clock, we have seen vortices emerge and cause intermingled defects in the presence of fixed coupling time-delays, so the main conclusions of the paper are not affected. However, the details of how delays could affect the system quantitatively are not yet worked out. This emergence of vortices is not surprising, as there are other systems with various forms of coupling time-delays that also show vortices (Jeong, Ko and Moon, 2002). In the manuscript, we now address the general issue of delays by citing these examples as an expectation that our conclusions should be robust to time-delays in the Discussion.

Reviewer #3:The manuscript by Uriu and colleagues addresses the process of resynchronization and segment recovery in somitogenesis in zebrafish embryos. On the one hand, it shows that recovery after early washout of DAPT drives the intermingling of defective and normal segments in zebrafish embryos. On the other hand, it presents an elegant yet complex mathematical and computational model to propose how this intermingling arises. The agreement between simulations and experiments is excellent. The manuscript is very well written, all data is very clear and the conclusions are very well supported. Videos are especially helpful to visualize the dynamics. I believe this is an excellent and elegant work that deserves publication in eLife.A major comments to improve the manuscript is:In the simulations, normal segments arise when the most posterior part of the tailbud spontaneously becomes synchronized. It is then when waves propagate from posterior to anterior, symmetrically on left and right sides, without any spiral. If I am correct, it is the geometry of the PSM and tailbud (two cylinders connected by a toroidal shape), together with the gradient of frequencies, what enables that "wave initiation" becomes ultimately localized within the wide region at the posterior end of the tailbud. During intermingling, wave initiation starts at other locations, more locally, within the PSM. I would suggest to discuss on the relevance of the overall geometry of the PSM+tailbud, that involves left-right symmetry, to reach re-synchronization. In the absence of left-right communication (e.g. in simulations where there is only one tubular shape with no toroidal mimicking the tailbud, for instance) I would expect that recovery of normal segmentation would take much longer, if it happens, and intermingling will be affected.

In reflecting on these comments, we realised that we had not explained the set up of the model properly with respect to how symmetrical patterns arise. Vortices are local disturbances in the phase pattern along the axis, but are still subject to the influence of the symmetrical frequency profile. Thus, in the absence of perturbations, the symmetrical wave pattern is in some sense the default. We also did not previously mention that the physical connection of both PSM sides at the tailbud is important for communication, as it ensures that the phase is the same on both sides of the posterior PSM.

We now include in the section on the physical model explicit statements about the left-right symmetry of the frequency profile and the corresponding symmetry of the wave pattern in the unperturbed state as well as the communication of phase through the tailbud. We also changed the section Simulations results 1 to contrast the symmetric recovered wave pattern and the asymmetries present in the perturbed, re-synchronizing pattern.

Using the thought experiment suggested by the reviewer, we hope this makes it clear that recovery of normal segmentation would take approximately the same time in a toroidal set-up compared to only one tube of the appropriate length, and intermingling will not be affected. The role of communication can be seen if we would compare two disconnected tubes. Then, due to small phase differences in the starting populations of cells at the posterior ends of these tubes, it is almost certain that the two tubes would develop patterns with identical shapes (given the same frequency profile is imposed), but some offset in the absolute phase values. This would give a correspondingly asymmetric placement of the segment boundaries when comparing the tubes, but – as above – there would be no difference in the recovery time or the intermingling.